# The *Rauvolfia tetraphylla* genome suggests multiple distinct biosynthetic routes for yohimbane monoterpene indole alkaloids

Emily Amor Stander[1,9], Beata Lehka[2,9], Inês Carqueijeiro [1,9], Clément Cuello [1,9], Frederik G. Hansson[2,9], Hans J. Jansen [3], Thomas Dugé De Bernonville[1,8], Caroline Birer Williams[1], Valentin Vergès[1], Enzo Lezin[1], Marcus Daniel Brandbjerg Bohn Lorensen[4], Thu-Thuy Dang [5], Audrey Oudin[1], Arnaud Lanoue[1], Mickael Durand [1], Nathalie Giglioli-Guivarc'h[1], Christian Janfelt[4], Nicolas Papon[6], Ron P. Dirks[3], Sarah Ellen O'connor [7✉], Michael Krogh Jensen [2✉], Sébastien Besseau [1✉] & Vincent Courdavault [1✉]

Monoterpene indole alkaloids (MIAs) are a structurally diverse family of specialized metabolites mainly produced in Gentianales to cope with environmental challenges. Due to their pharmacological properties, the biosynthetic modalities of several MIA types have been elucidated but not that of the yohimbanes. Here, we combine metabolomics, proteomics, transcriptomics and genome sequencing of *Rauvolfia tetraphylla* with machine learning to discover the unexpected multiple actors of this natural product synthesis. We identify a medium chain dehydrogenase/ reductase (MDR) that produces a mixture of four diastereomers of yohimbanes including the well-known yohimbine and rauwolscine. In addition to this multifunctional yohimbane synthase (YOS), an MDR synthesizing mainly heteroyohimbanes and the short chain dehydrogenase vitrosamine synthase also display a yohimbane synthase side activity. Lastly, we establish that the combination of geissoschizine synthase with at least three other MDRs also produces a yohimbane mixture thus shedding light on the complex mechanisms evolved for the synthesis of these plant bioactives.

[1] Biomolécules et Biotechnologies Végétales, EA2106, Université de Tours, 37200 Tours, France. [2] Novo Nordisk Foundation Center for Biosustainability, Technical University of Denmark, Kgs, Lyngby, Denmark. [3] Future Genomics Technologies, 2333 BE, Leiden, The Netherlands. [4] Department of Pharmacy, Faculty of Health and Medical Sciences, University of Copenhagen, Universitetsparken 2, 2100 Copenhagen, Denmark. [5] Department of Chemistry, Irving K. Barber Faculty of Science, University of British Columbia, Kelowna, BC, Canada. [6] Univ Angers, Univ Brest, IRF, SFR ICAT, F-49000 Angers, France. [7] Department of Natural Product Biosynthesis, Max Planck Institute for Chemical Ecology, Jena 07745, Germany. [8]Present address: Limagrain, Centre de Recherche, Route d'Ennezat, Chappes, France. [9]These authors contributed equally: Emily Amor Stander, Beata Lehka, Inês Carqueijeiro, Clément Cuello, Frederik G. Hansson. ✉email: oconnor@ice.mpg.de; mije@dtu.dk; sebastien.besseau@univ-tours.fr; vincent.courdavault@univ-tours.fr

Specialized metabolites have evolved in plants to cope with environmental pressure and the myriad of biotic and abiotic stresses imposed by their sessile status. Within this broad group of metabolites, alkaloids are compounds with diverse physiological roles and often described as highly poisonous molecules that protect plants against pests and herbivores. The monoterpene indole alkaloids (MIAs) are one of the most structurally diverse families of alkaloids, with more than 2500 distinct molecules identified so far. MIAs are widespread within the Gentianales order including Apocynaceae, Gelsemiaceae, Loganiaceae, and Rubiaceae, and are also biosynthesized in Nyssaceae (Cornales order) and in Icacinaceae[1]. Almost all MIAs derive from a common precursor, strictosidine, which undergoes a series of complex rearrangements leading to the formation of distinct main MIA classes, based on the skeleton configuration[2,3]. These include the sarpagan, yohimbane and heteroyohimbane, aspidosperma, iboga or quinoline MIA types that are heterogeneously distributed among the MIA producing plants[4]. While the ecological functions of most MIAs remain cryptic, recent work suggests that these compounds protect the producing plants against certain herbivores[5]. Numerous MIAs also exhibit astonishing pharmaceutical properties encompassing the anticancer vinblastine and vincristine, the antimalarial quinine, the antihypertensive ajmalicine or the antiarrhythmic ajmaline that explain the ever-growing interest of the scientific community for the study of their biosynthesis. In addition, many MIAs also display undervalued or underexploited biological activities due to their low bioavailability *in planta* while other MIAs have recreational uses such as the yohimbane MIA yohimbine described as a stimulant.

Over years, characterization of the MIA biosynthesis has been mainly performed through the combination of metabolomic, biochemical and transcriptomic approaches performed using the iconic MIA producing plant *Catharanthus roseus* and to a lesser extent in *Rauvolfia* sp.[6–9]. Recently, the sequencing of many MIA producing plant genomes have provided complementary unprecedented insights into the understanding of their metabolism[10]. These genomes include *Rhazya stricta*[11], *Gelsemium sempervirens*[12], *Camptotheca accuminata*[13], *Ophiorrhiza pumila*[14], *Mitragyna speciosa*[15], *Neolamarckia cadamba*[16], *Vinca minor*[17], *Voacanga thouarsii*[18] and *C. roseus*[12,19–22]. To date, all MIA biosynthetic steps up to the strictosidine biosynthetic intermediate have been discovered. This includes the upstream enzymes responsible for the transformation of geranyl diphosphate into secologanin, the ultimate terpenoid precursor of MIAs, but also the tryptophan decarboxylase (TDC) catalyzing the decarboxylation of tryptophan into tryptamine, the indole MIA precursor. Secologanin and tryptamine are condensed by strictosidine synthase (STR) to from strictosidine, which is then deglucosylated by strictosidine β-D-glucosidase (SGD) that has been cloned in both *C. roseus* and *R. serpentina*[23,24]. The deglucosylation of strictosidine leads to the formation of a highly reactive aglycone that causes protein reticulation as a part of a defense response[25,26]. Notably, strictosidine aglycone also undergoes spontaneous rearrangements to form 4,21-dehydrogeissoschizine, which is then reduced by dedicated alcohol dehydrogenases (ADHs). These reduced products are then shuttled into various pathway branches that eventually generate the plethora of downstream MIAs, confirming the major contribution of ADHs to MIA structural diversity[27].

Indeed, the medium chain dehydrogenase/reductase (MDR) geissoschizine synthase (GS) reduces 4,21-dehydrogeissoschizine into geissoschizine, which enters the biosynthetic pathway of sarpagan, aspidosperma and iboga MIAs including among others tabersonine/catharanthine in *C. roseus*[28–30], ajmaline in *Rauvolfia* sp.[31], ibogaine in *Tabernanthe iboga*[32] or strychnine in *Strychnos nux vomica*[33]. The 4,21-dehydrogeissoschizine can also undergo spontaneous rearrangements to form cathenamine and 19-epi-cathenamine whose reduction through chemical treatments or co-incubation with *C. roseus* crude protein extracts with NAPDH yields the formation of heteroyohimbane MIAs such as ajmalicine or tetrahydroalstonine[34,35]. Five MDRs that reduce cathenamine/epi-cathenamine have been then characterized in *C. roseus* including four tetrahydroalstonine synthases (THAS1-THAS4) and heteroyohimbine synthase (HYS), which synthesizes a mixture of ajmalicine, tetrahydroalstonine and 19-epi-ajmalicine[36,37]. Interestingly, THAS1 and THAS2 also reduce the iminium precursor of vinblastine/vincristine[21]. A short-chain alcohol dehydrogenase (SDR), namely vitrosamine synthase (VAS) has also been described to reduce the strictosidine aglycone into the cryptic MIA vitrosamine (VAS)[38]. In addition, members of the aldo-keto reductase (AKR) family have been recruited for MIA biosynthesis, including redox2 that reduces akuammicine derivatives in *C. roseus* and can work in combination with GS to produce isositsirikine[30]. This also includes the perakine reductase (PR) that ensures the reduction of the sarpagan MIA perakine into raucaffrinolide in *R. serpentina*[39]. However, no enzyme involved in the biosynthesis of yohimbanes including the four isomers yohimbine, rauwolscine, corynanthine, and allo-yohimbine has been identified to date (Fig. 1). Interestingly, rauwolscine, also known as α-yohimbine, and yohimbine are described as specific antagonists of brain α2-adrenergic receptors that can be used, respectively, to treat levodopa-induced dyskinesia in Parkinson's disease and as a tool to study alcohol use disorder[40–42]. Allo-yohimbine displays similar pharmacological properties but corynanthine is known to be a selective α1-adrenergic receptor antagonist reducing intraocular pressure for instance[43]. While these MIAs are synthesized in many plants, rauwolscine and yohimbine are highly accumulated in *Rauvolfia* sp. and notably in *R. tetraphylla* ("be still tree" or "devil-pepper") making of this plant a powerful model system for the elucidation of yohimbane biosynthesis (Fig. 2a)[44–47].

Here, we combine transcriptomics, genome sequencing in *R. tetraphylla* to identify biosynthetic genes for yohimbanes using a machine learning (ML) approach. We established that many of the ADH encoding genes were clustered in the genome and displayed expression patterns similar to other previously characterized MIA biosynthetic genes. Secondly, by testing enzyme activity of the identified genes in yeast and using recombinant proteins, we identified three enzymes that produce yohimbanes both in vitro and in vivo, when assayed with SGD and strictosidine, including a highly efficient enzyme producing several yohimbane isomers. Finally, we also demonstrated that these MIAs can be also produced through a double-enzyme mechanism including a GS ortholog thus highlighting the diversity of yohimbane biosynthetic routes evolved by *R. tetraphylla*.

## Results

**Genome sequencing and assembly of *R. tetraphylla*.** After DNA extraction from young leaves and sequencing, the *R. tetraphylla* genome was first assembled into 1008 contigs with an N50 of 3.7 Mb. After haplotigs removal and a final pilon polishing, the 364,945,498 bp final assembly was distributed across 76 scaffolds with an N50 of 8.135 Mb and a GC content of 33.89% (Supplementary Table S1, Fig. 3a, b). The base level QV of 32.6546, corresponding to more than 99,999% base accuracy, together with the LTR assembly index of 18.90 are a very good indicators of the high quality of the assembled genome. The assembled genome reached a complete BUSCO score of 96.2% based on core *Eudicotyledons* genes (Fig. 2b) thus highlighting its completeness.

**Gene annotation and transposable element discovery in *R. tetraphylla*.** Gene annotation was performed through RNAseq-based gene modeling and resulted in the identification of 23,228

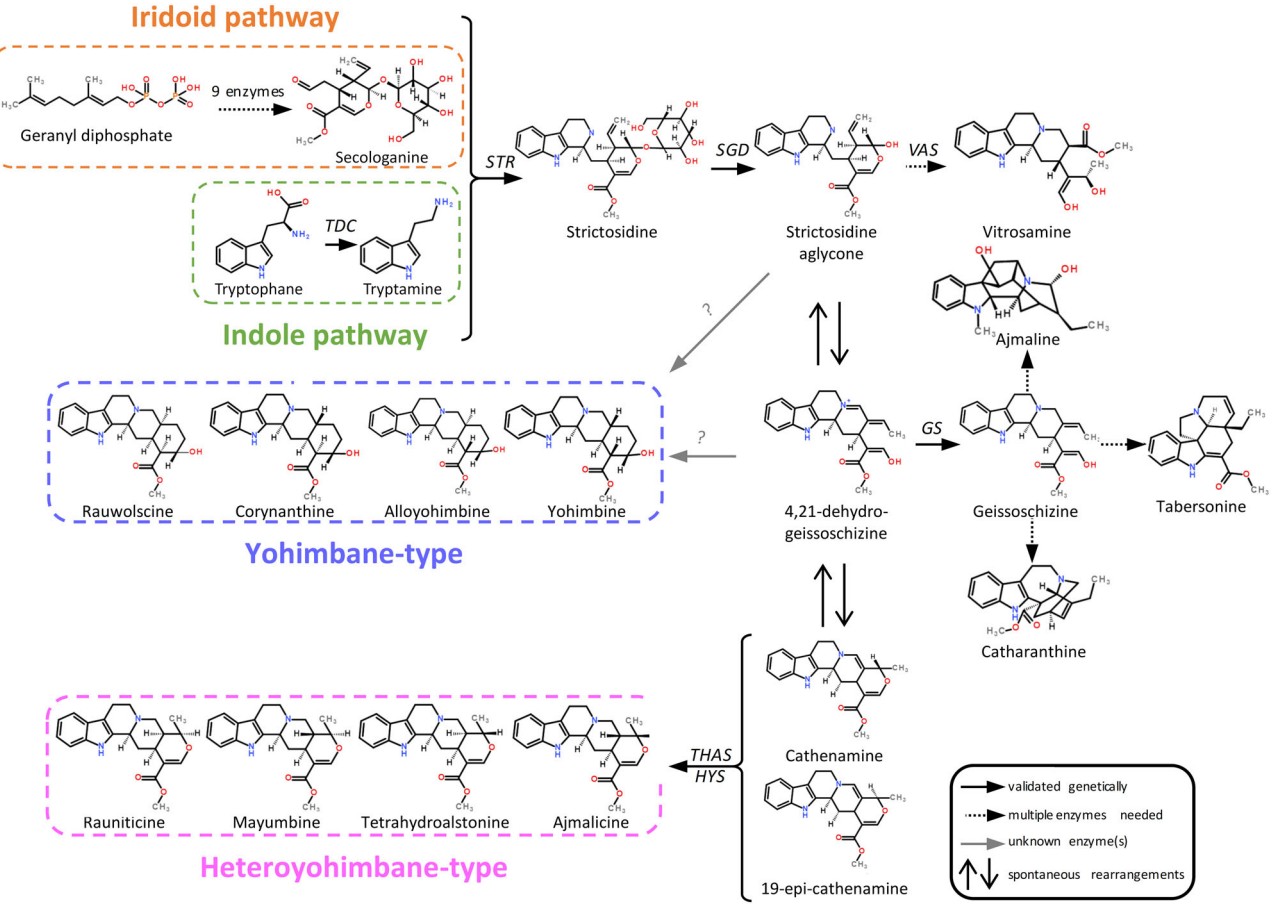

**Fig. 1 Proposed biosynthesis pathway of yohimbane (blue) and heteroyohimbane (pink) monoterpene indole alkaloids (MIAs).** MIA biosynthesis starts with the condensation of secologanin, the monoterpene precursor derived from geranyl diphosphate, with tryptamine, the indole precursor generated through the decarboxylation of tryptophan catalyzed by tryptophan decarboxylase (TDC). This reaction performed by strictosidine synthase (STR) produces the universal MIA precursor strictosidine, which is subsequently deglycosylated by strictosidine β-D-glucosidase (SGD). This aglycone intermediate is then used for downstream MIA biosynthesis including the synthesis of vitrosamine by the vitrosamine synthase (VAS). The strictosidine aglycone can also spontaneously rearrange into the less reactive 4,21 dehydrogeissoschizine that can be used as substrate by geissoschizine synthase (GS) to produce geissoschizine, known as the precursor of the iboga-type catharanthine, the aspidosperma-type tabersonine or the sarpargan-type ajmaline. 4,21 dehydrogeissoschizine is also proposed to serve as a precursor of the yohimbane MIAs including the four isomers rauwolscine, corynanthine, alloyohimbine, and yohimbine, whose biosynthetic modalities remain uncharacterized. Lastly, 4,21 dehydrogeissoschizine can also spontaneously cyclised to from cathenamine and 19-epi-cathenamine that are converted by tetrahydroalstonine synthase (THAS) and heteroyohimbine synthase (HYS) into several heteroyohimbane MIA isomers including tetrahydroalstonine, ajmalicine, mayumbine, and rauniticine.

protein coding genes, which is comparable to previously characterized Apocynaceae species[17,18,20,48]. The *Eudicotyledons* BUSCO score of 93.9% for this gene set indicates good completeness (Fig. 2b). For almost all large scaffolds, a high gene density was observed (Fig. 3c).

The combination of BLASTX against the UniProt database and hmmscan against the PFAM database led to the functional annotation of 80.3% (18,658 / 23,228 genes) of the predicted genes (Supplementary Data S1). More than half of the genes (55.6%, 12,924 / 23,228 genes) were annotated by all four databases. Putative orthologs of functionally validated MIA biosynthetic genes were identified using BLASTP against protein sequences from *C. roseus*, *V. minor*, *T. iboga*, *G. sempervirens*, and *Rauvolfia* sp. by considering hits with at least 90% coverage and 60% identity (Supplementary Data S2). Notably, high confidence orthologs (82–96% protein identity) for all genes of the MIA terpene precursor and the MIA biosynthetic pathway up to geissoschizine were identified. We observed that these MIA genes were heterogeneously distributed among genome scaffolds with enriched regions (Fig. 3d). In contrast, genes predicted to encode post-geissoschizine steps

were of a lower identity, given the high MIA diversification across species, except orthologs for *THAS* notably (85–98% protein identity). Among the putative ADH genes, 79 MDR (PF00107.26, PF08240.12), 76 SDR (PF00106.25) and 27 AKR (PF00248.21) were identified in the *R. tetraphylla* genome (Supplementary Fig. S1) suggesting a slight increase of these families compared to other Apocynaceae (Supplementary Table S2). The majority of ADH encoding genes was distributed in all genomic scaffolds (Fig. 3e).

Finally, given the major roles of transposable elements (TE) in genome evolution, genetic instability and gene expression regulation[49], the TE composition of the *R. tetraphylla* genome was also studied. Interestingly, 43.27% of the genome consists of transposable elements, mainly long-terminal repeat retrotransposons from the Copia family (Fig. 3f, Supplementary Table S3). This proportion of TE thus appears very similar to that of the closely related species *C. roseus* (42.87%[20]).

**Gene duplication and evolutionary divergence of *R. tetraphylla*.** Whole genome duplication (WGD) events are one of the major diversification mechanisms of plants[10]. Thus, to infer WGD

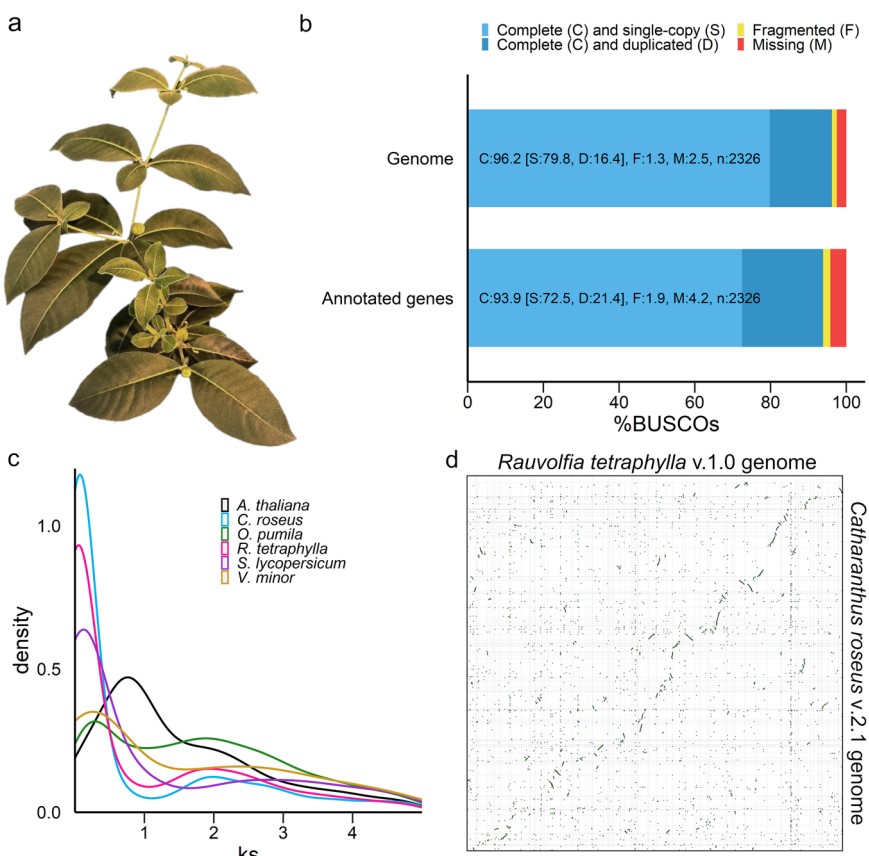

**Fig. 2 General characteristics of the R. tetraphylla genome. a** *R. tetraphylla*, commonly known as be still tree or devil-pepper, is a small, evergreen shrub native to central America. **b** BUSCO scores of genome and annotated genes. **c** Synonymous substitution rate (Ks) distribution plot for *R. tetraphylla* paralogs compared to other *Eudicotyledons*. **d** Genome-wide synteny with *C. roseus* v2.1 genome[20].

events in *R. tetraphylla*, we identified paralogous gene pairs across eleven species including nine MIA-producing species from the Apocynaceae (*C. roseus, R. tetraphylla, V. minor, V. thouarsii, C. gigantea*), Gelsemiaceae (*G. sempervirens*), Rubiaceae (*O. pumila, M. speciosa*), and Nyssaceae (*C. acuminata*) families, as well as two non-MIA-producing species (*A. thaliana, S. lycopersicum*) and calculated the synonymous substitution per synonymous site (Ks) for each gene pair (Fig. 2c, Supplementary Data S3). Ks distribution clearly shows one Ks peak at around Ks = 2 corresponding to the ancient γ whole-genome triplication shared among all *Eudicotyledons* with no other duplication events.

A genome-wide synteny analysis between *R. tetraphylla* and the well-known closely related species *C. roseus* was carried out. While it can be somewhat hidden by both genome fragmentation, the resulting dot plot clearly showed an evident collinearity between the two genomes (Fig. 2d).

To gain further insight into *R. tetraphylla* genome evolution, we generated gene families (orthogroups) across the same eleven species 92.4% of genes (272,661 / 295,045) were assigned to 23,480 orthogroups across all species, with a mean orthogroup size of 11.6 proteins. 645 single-copy orthogroups were used to build a maximum-likelihood phylogenetic tree (Fig. 4a). We investigated gene family evolution across the obtained phylogenetic tree. Overall, *R. tetraphylla* genome showed an increase of 2054 orthogroups and a decrease of 889 orthogroups (Fig. 4a). A total of 174 biological processes, 44 cellular component and 177 molecular function GO terms were significantly enriched in *R. tetraphylla* increased orthogroups (Supplementary Data S4). Interestingly, enriched biological processes included response to diverse stimuli (Fig. 4b) including abiotic stresses (light: GO:0009650, GO:0010018,

GO:0070914, GO:0071493 ; drought: GO:0080148, GO:0009819 ; temperature: GO:0009408, GO:0010378), biotic stresses (insects: GO:0009625 ; fungi : GO:0050832) ; endogenous stimuli (hormones: GO:0009788, GO:0009861, GO:0071367, GO:0009753, GO:0009734, GO:0009787, GO:0080026) and external stimuli (chemicals: GO:0071291, GO:0035874, GO:0006805 ; nutrients: GO:0090549, GO:0080029 ; toxic substances: GO:0098754) ; and specialized metabolism (Fig. 4c) including secondary metabolic processes (GO:0019748), flavonoid metabolic processes (GO:0009813, GO:0009812) and most interestingly indole-containing compound metabolism processes (GO:0000162, GO:0009759). Indeed, 10 out of the 25 genes associated with tryptophan biosynthesis (*anthranilate synthase*-like: MSTRG.10553, MSTRG.17213, MSTRG.313 ; *anthranilate phosphoribosyltransferase*-like: MSTRG.13694, MSTRG.17230 ; N-(5'-phosphoribosyl) anthranilate isomerase 1-like: MSTRG.10092 ; *indole-3-glycerol phosphate synthase*-like: MSTRG.17336 ; CYP83B1-like: MSTRG.11866, MSTRG.11867, MSTRG.11869) and 3 of the 5 genes associated with indole glucosinolate biosynthesis (*CYP83B1*-like: MSTRG.11866, MSTRG.11867, MSTRG.11869) were present in *R. tetraphylla* extended orthogroups. Such evolutionary processes may account for the specialized metabolite variability found in closely related MIA-producing species, such as *R. tetraphylla* and *C. roseus*.

**Creating an integrated MIA and gene expression atlas for MIA biosynthetic gene discovery.** To initiate the quest for enzymes synthesizing yohimbanes, a global atlas of gene expression levels was prepared by sampling 16 *R. tetraphylla* tissue types or

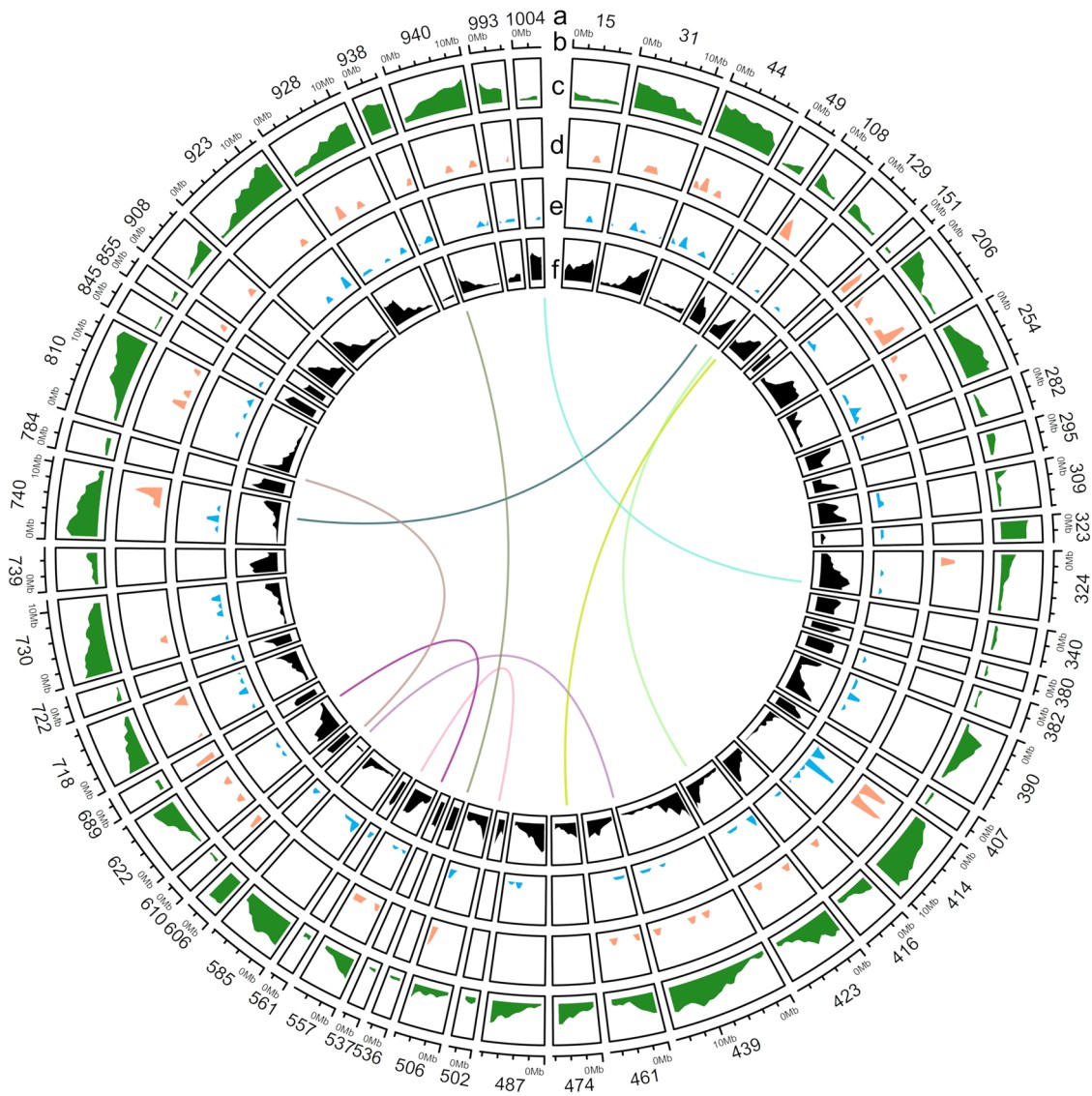

**Fig. 3 Genomic landscape of _R. tetraphylla_.** Only scaffolds of at least 2 Mb are represented. Concentric rings present scaffold name (**a**), scaffold scale (**b**), gene density (**c**), ortholog of alkaloid biosynthetic genes position (**d**), alcohol dehydrogenase density (**e**) and Copia density (**f**). Central links represent collinearity.

experimental conditions for RNA extraction (Fig. 5a, Supplementary Data S5). We also took advantage of the ability of _M. sexta_ caterpillars to feed on leaves to create libraries of biotically stressed leaf material as originally described for _C. roseus_[5]. A multi-dimensional scaling was conducted to visualize the full samples and evaluate sample reproducibility (Supplementary Fig. S2). Replicates clustered well together (Pearson correlation = 0.91+/− 0.05 over the 16 tissue types). Root tissues were clearly separated from the other in the first dimension. The second dimension more likely distinguished young leaf tissues (seedling) from the old ones. Other tissue types such as berries, flowers and stems were not resolved in these two dimensions, showing they are characterized by specific gene expression profiles. According to these results, our dataset displays a strong variability in the gene expression levels across the different tissue types, thereby creating a powerful resource for gene co-expression studies.

In addition, we performed an extensive MIA quantification on the same _R. tetraphylla_ samples (Supplementary Data S6). Notably, this confirmed the presence of high amounts of reserpilline and isoreserpilline in young, old leaves and flowers as observed in other _Rauvolfia_ sp[47]. (Fig. 5b). Heteroyohimbanes were also well

represented with THA and ajmalicine highly accumulated in flowers and young leaves, respectively. We noted that their oxidized derivatives, namely alstonine and serpentine were preferentially accumulated in stems and roots. Interestingly, yohimbane distribution was slightly different with rauwolscine being the most accumulated in many of the tested organs except berries and roots (Fig. 5c). Corynanthine and yohimbine were present at lower levels and preferentially in young leaves. Finally, we also observed that leaf consumption by _M. sexta_ induces MIA metabolism (Fig. 5d, e).

**Detection of genes encoding MIA-related ADHs involve in yohimbane synthesis via gene coexpression analysis and machine learning.** Based on the most probable biosynthetic scenario, we initiated the prediction of candidate genes involved in yohimbane synthesis by focusing on ADHs given their already established capacity of cyclising geissoschizine to produce corynanthe MIAs[36,37]. Predicting these candidates was achieved by the combination of distinct approaches starting with a guilty-by-association strategy. The RNA-seq compendium was thus mined for gene expression levels using a classical co-expression network. A total of

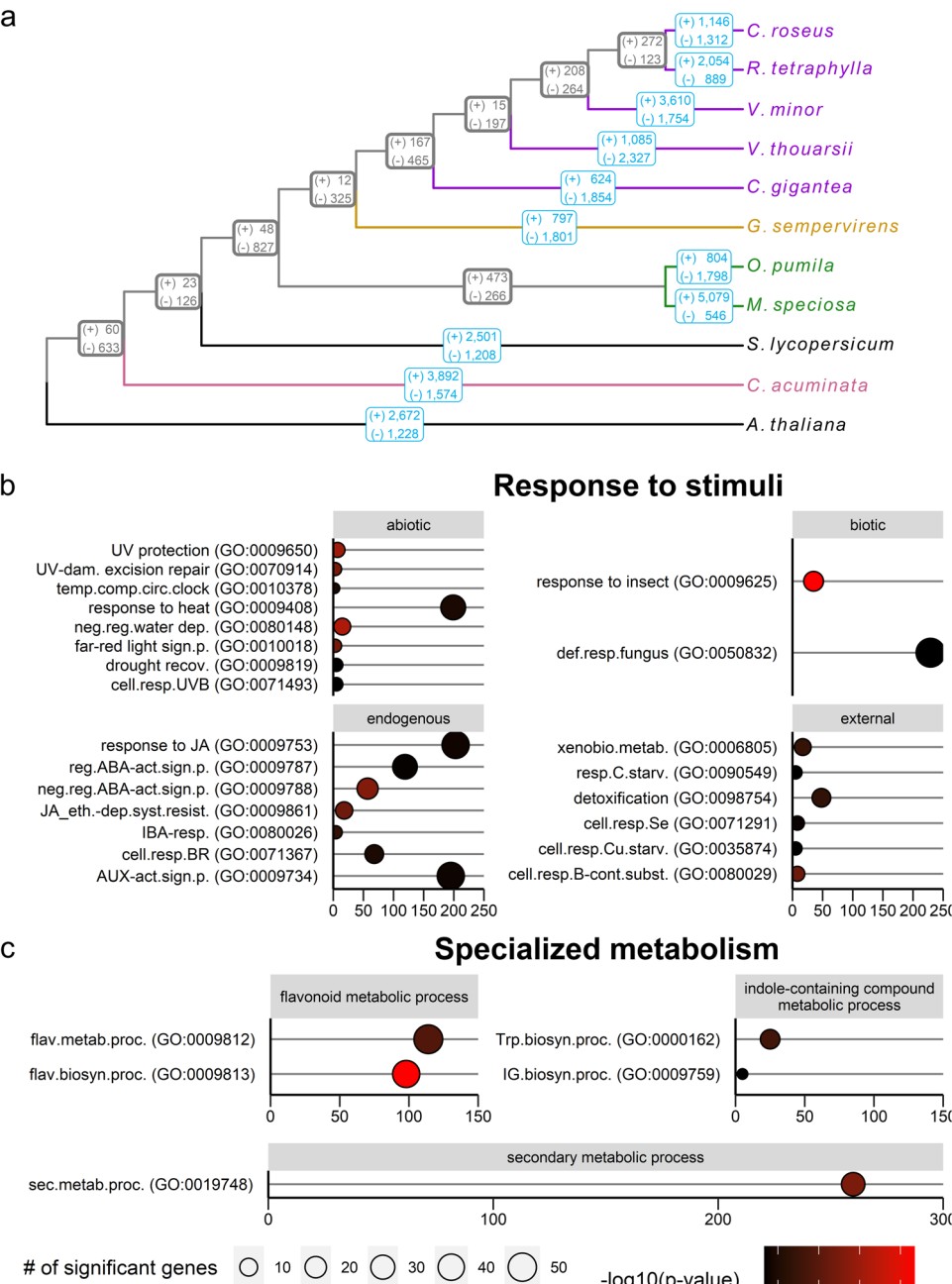

**Fig. 4 Comparative genomic analysis of _R. tetraphylla_ with 10 other plant species. a** Orthofinder phylogenetic tree. Colored branches represent MIA-producing plant families including five Apocynaceae (purple: _C. roseus_, _R. tetraphylla_, _V. minor_, _V. thouarsii_, _C. gigantea_), one Gelsemiaceae (yellow: _G. sempervirens_), two Rubiaceae (green: _O. pumila_, _M. speciosa_) and one Cornales (pink: _C. acuminata_). Gene family size changes were calculated with Cafe5[87]. Light border boxes: expanded (+) or contracted (−) gene families in each lineage, thick bordered boxes: expanded (+) or contracted (−) gene families in internal nodes of ancestral populations for each taxon. **b**, **c** Enriched biological processes in _R. tetraphylla_ expanded orthogroups associated with response to stimuli (**b**, 23 terms) and specialized metabolism (**c**, 5 terms). y-axis: GO terms, x-axis: number of genes in each GO term, circle size: number of significant genes in each GO term, fill color: significance level (Fisher's exact test), neg.reg.water dep.: negative regulation of response to water deprivation, UV protection: UV protection, far-red light sign.p.: far-red light signaling pathway, UV-dam. excision repair: UV-damage excision repair, temp.comp.circ.clock: temperature compensation of the circadian clock, response to heat: response to heat, drought recov.: drought recovery, cell.resp.UVB: cellular response to UV-B, response to insect: response to insect, def.resp.fungus: defense response to fungus, neg.reg.ABA-act.sign.p.: negative regulation of abscisic acid-activated signaling pathway, JA_eth.-dep.syst.resist.: jasmonic acid and ethylene-dependent systemic resistanc, IBA-resp.: response to indolebutyric acid, cell.resp.BR: cellular response to brassinosteroid stimulus, response to JA: response to jasmonic acid, AUX-act.sign.p.: auxin-activated signaling pathway, reg.ABA-act.sign.p.: regulation of abscisic acid-activated signaling pathway, cell.resp.B-cont.subst.: cellular response to boron-containing substance levels, xenobio.metab.: xenobiotic metabolic process, detoxification: detoxification, cell.resp.Se: cellular response to selenium ion, cell.resp.Cu.starv.: cellular response to copper ion starvation, resp.C.starv.: response to carbon starvation, flav.biosyn.proc.: flavonoid biosynthetic process, sec.metab.proc.: secondary metabolic process, flav.metab.proc.: flavonoid metabolic process, Trp.biosyn.proc.: tryptophan biosynthetic process, IG.biosyn.proc.: indole glucosinolate biosynthetic process.

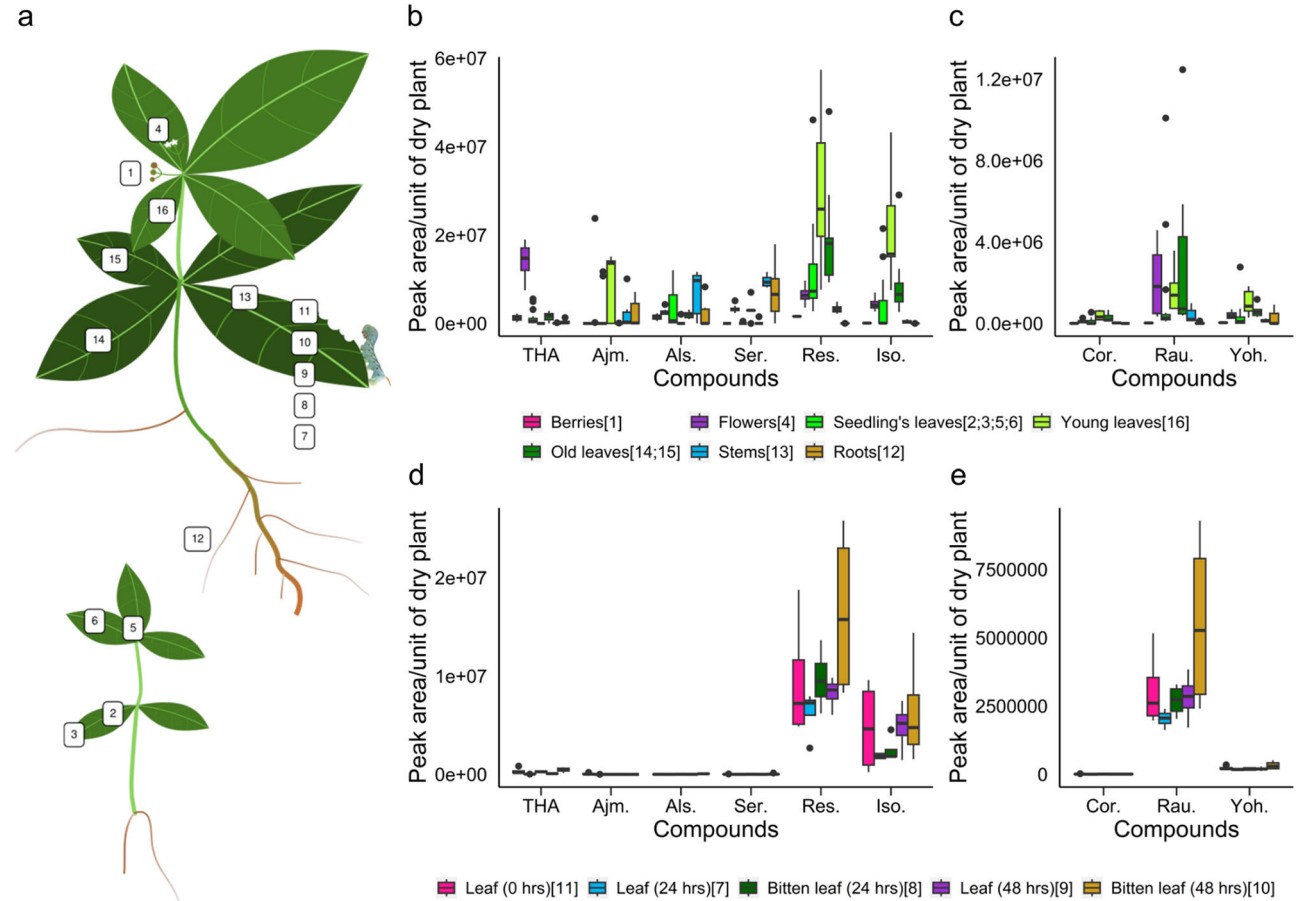

**Fig. 5 Organs sampling an MIA quantification in *R. tetraphylla*. a** Overview of the gene expression atlas samples with numbers surrounded by boxes representing tissue type (Supplementary Data S4). **b**, **c**, **d**, **e** Relative alkaloid quantification of the samples from the *R. tetraphylla* gene expression atlas, with tetrahydroalstonine (THA), ajmalicine (Ajm.), alstonine (Als.), serpentine (Ser.), reserpiline (Res.), isoreserpiline (Iso.) and the three yohimbanes corynanthine (Cor.), rauwolscine (Rau.) and yohimbine (Yoh.) in different organs/developmental stages (**b**, **c**) and in leaves attacked by *M. sexta* (**d**, **e**). $n = 7$ biologically independent samples.

558 transcripts were found in the coexpression network, including 64 with close homology to known MIA genes and 32 ADHs encoding transcripts annotated as MDR (PF00107.26; PF08240.12), SDR (PF00106.25) or AKR (PF00248.21) (Supplementary Data S7). The resulting coexpression network clearly shows some tightly connected communities (Fig. 6a). Five out of the 19 MDR encoding transcripts (*MSTRG.14221.1*, *MSTRG.5289.1*, *MSTRG.5293.1*, *MSTRG.5294.1* and *MSTRG.5528.1*) had weaker similarities to known MIA genes (<90% identity), suggesting they may catalyze still unknown enzymatic reactions in the MIA pathways. Two had strong similarities with known MIA related genes (*MSTRG.5284.1* 98% id with *VR* and *MSTRG.5290.1* 95% id with *THAS1*).

These first candidates were then completed by prediction based on a feedforward artificial neural network. A simple deep learning network was thus trained to classify transcripts as MIA- or non-MIA-related as described for the identification of missing steps in the tropane alkaloid metabolism[50]. Despite the definition of true positives (best blast hits for known MIA genes), it was less obvious to select a proper set of true negatives (non-MIA-related). We used genes predicted as conserved genes using the BUSCO methods. We finally trained a model able to correctly predict 41 transcripts as MIA related (out of 75 annotated by BLAST). The Area Under Curve (AUC) on the validation datasets and cross-validation was always lower than that of the training datasets, while still retaining a good performance (Fig. 6b, c, Supplementary Table S4). Besides, the model predicted 485 transcripts (initially unlabeled) as

potential MIA-related sequences because they shared similarities in their accumulation pattern (Supplementary Data S8). Inspection of the newly labeled MIA sequences revealed the presence of 28 transcripts encoding putative ADHs including 9 new candidates (*MSTRG.2694.1*, *MSTRG.5283.1*, *MSTRG.5538.5*, *MSTRG.5530.2*, *MSTRG.12039.1*, *MSTRG.7057.1*, *MSTRG.9273.2*, *MSTRG.5533.4*, *MSTRG.9264.2*) compared to the co-expression network analysis. Among MIA encoding transcripts, one had strong homologies with the *PR* transcript (*MSTRG.20447.1* 95% id). The other candidates had about 85% id *8HGO* (*MSTRG.12039.1* and *MSTRG.14221.1*)[51], *THAS* (*MSTRG.5294.1*)[36,37], and *GS* (*MSTRG.5528.1*)[28].

**Prediction of yohimbane synthase candidate by the proteomic analysis of the *R. tetraphylla* latex.** In a second step, we searched for yohimbane synthase candidates via a more targeted approach. From the metabolite profiling of *R. tetraphylla* tissues, we identified rauwolscine as a major MIA from the latex exuding from leaves, besides alstonine (Fig. 7a, b, Supplementary Data S6). This suggested the potential presence of ADHs displaying a rauwolscine synthase activity in latex, thus prompting us to determine its protein content. The proteomic analysis of the latex revealed a total of 575 unique proteins, 550 of which were annotated via the Swissprot database (Fig. 7d, Supplementary Data S9). In addition to the expected highly accumulated major latex proteins, we found 9 putative MDR, 10 SDR and 3 AKR within 92 proteins with GO

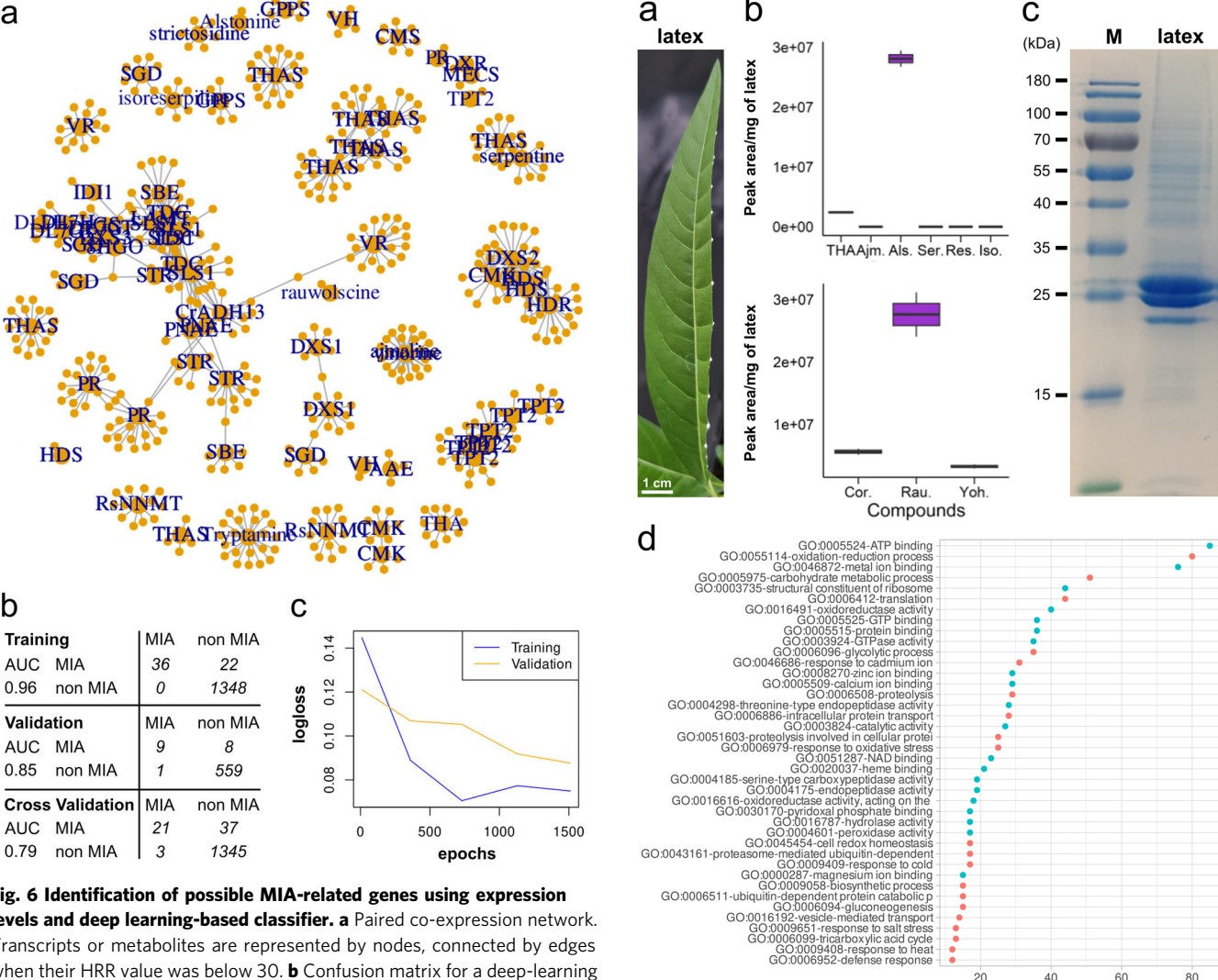

**Fig. 6 Identification of possible MIA-related genes using expression levels and deep learning-based classifier. a** Paired co-expression network. Transcripts or metabolites are represented by nodes, connected by edges when their HRR value was below 30. **b** Confusion matrix for a deep-learning based identification of MIA-related genes. The initial expression matrix was split into one training and one validation dataset and the deep learning network was constructed with a 5-fold cross validation. The Area Under the Curve (AUC) is indicated for each dataset. Numbers correspond to the number of transcripts falling into each class predicted or already known. **c** Evolution of the logloss as the training and validation datasets are passing through the network (epochs).

**Fig. 7 Proteomics and metabolomics analysis of the *R. tetraphylla* latex.** Latex from cutted leaves (**a**) has been collected for MIA quantification (**b**, *n* = 3 biologically independent samples) and analysis of its protein content by SDS-PAGE (**c**). Enriched biological processes of the sequenced proteins of the *R. tetraphylla* latex (**d**). Red and blue dots correspond to GO terms from the biological process and molecular function respectively.

term associated to reduction process such as GO:0055114 (oxidation-reduction process) or GO:0016491 (oxidoreductase activity). (Fig. 7d) (Supplementary Data S9). This notably includes the ortholog of the PR from *R. serpentina* (*MSTRG.20447.1*[38]) but also two MDRs (*MSTRG.5283* and *MSTRG.5534*) displaying homology with the THAS previously identified in *C. roseus*[36,37].

**MIA gene clusters encompassing putative yohimbane synthase candidates.** Finally, we searched for yohimbane synthase candidates by seeking for physically co-localized metabolic genes of interest by anchoring the predicted MIA orthologs onto the *R. tetraphylla* genome and examining the surrounding genomic regions for identifying putative metabolic genes of interest. This analysis revealed 76 putative MIA gene clusters (Supplementary Data S10) including 37 MDR and 7 SDR (Supplementary Data S11). Two clusters containing *STR* and *TDC* (cluster 32 and cluster 55), which have been previously described in *C. roseus*, *G. sempervirens*, *R. stricta*, and *V. thouarsii*[11,12,18,19,21,22]. We were also able to identify other

biosynthetic gene clusters described in the closely related species *C. roseus*[21,22] including *TAT/DAT* (cluster 29), *16OMT/T3R/DL7H* (cluster 20), *Redox1/10HGO/ADH9/GS* (cluster 21) and *SAT/T3O* (cluster 8). Conversely, some clusters, notably the well described *T16H/16OMT*[12,17,21,22], could not be found. As the orthologs of these genes are not found on the same contig, it is possible that this cluster exists in *R. tetraphylla* but that the present genome is still too fragmented to observe this. Both recent chromosome-scale *C. roseus* genomes[21,22] highlighted THAS and HYS clusters resulting from local duplications. Interestingly, we also found two genomic regions highly enriched in ADH encoding genes both being located on contig 414 (Fig. 8). Indeed, cluster 30 comprised two MIA gene orthologs (*MSTRG.5280*: stemmadenine *O*-acetyltransferase, *MSTRG.5282*: *O*-acetylstemmadenine oxidase), nine putatively annotated ADH genes including seven candidates identified previously (*MSTRG.5283*, *MSTRG.5284*, *MSTRG.5287*, *MSTRG.5288*, *MSTRG.5289*, *MSTRG.5290*, *MSTRG.5293*, *MSTRG.5294*) and seven putatively annotated cytochromes P450. Cluster 31 is

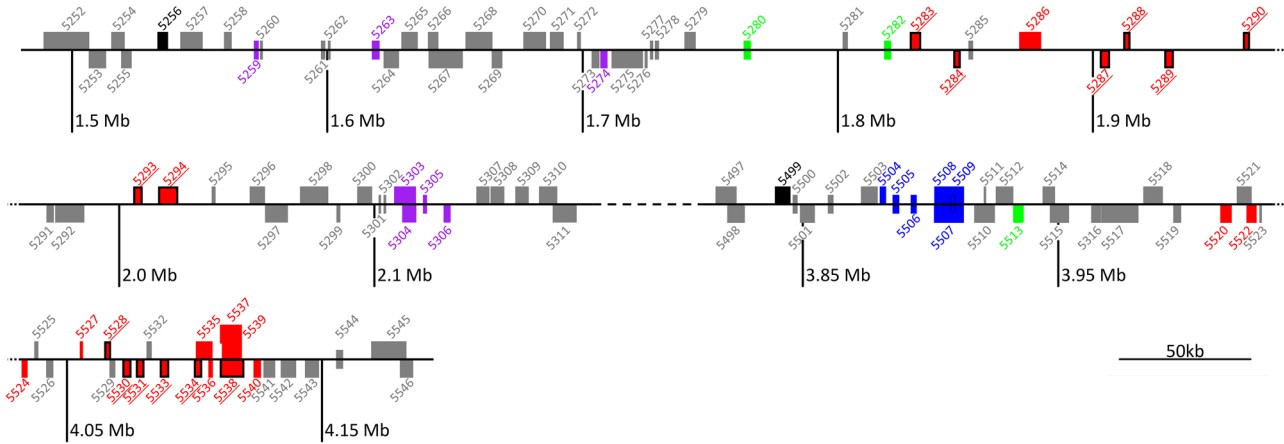

**Fig. 8 ADH-rich gene clusters on contig 414 of the *R. tetraphylla* genome.** Red boxes: putative alcohol dehydrogenases, black bordered red boxes: ADH candidates, purple boxes: cytochrome P450, blue boxes: UDP-glycosyltransferases, green boxes: MIA enzyme orthologs (MSTRG.5280: stemmadenine *O*-acetyltransferase-like, MSTRG.5282: *O*-acetylstemmadenine oxidase-like, MSTRG.5513: 1-deoxy-D-xylulose-5-phosphate synthase-like), black boxes: transcription factors, gray boxes: other functions.

composed of one MIA gene ortholog (*MSTRG.5513*: 1-deoxy-D-xylulose-5-phosphate synthase), 15 putative ADHs including six candidates (*MSTRG.5528, MSTRG.5530, MSTRG.5531, MSTRG.5533, MSTRG.5534, MSTRG.5538*) downstream from six putatively annotated UDP-glycosyltransferases. Interestingly, these two clusters show a strong synteny with a genomic region from *C. roseus* (contig 884 of *C. roseus* v.2.1; Supplementary Fig. S3) and probably is a result of an ancient tandem or proximal duplication event.

**The functional validation of ADH candidates led to the identification of a yohimbane synthase producing four yohimbane isomers.** To prioritize potential yohimbane synthase candidates, all predicted ADHs generated through coexpression networks (27; Fig. 6a), machine learning (20 ; Fig. 6b, c), latex proteomics (19 ; Fig. 7a–c ; Supplementary Fig. S4) and gene cluster analysis (44; Fig. 8) were combined and duplicated ADHs from different methods as well as incomplete (partial sequence, absence of ATG or stop codons) and alternative sequences were removed. On this basis, 61 MIA-related ADHs gene candidates were retained including 21 SDR, 5 AKR and 35 MDR (Supplementary Data S12). To initiate functional validation, we first focused on this last ADH subfamily given the preponderant role of MDR in MIA biosynthesis. 14 putative candidates identified from at least 2 different prediction approaches were selected (*MSTRG.747; MSTRG.2694; MSTRG.5283; MSTRG.5293; MSTRG.5294; MSTRG.5534; MSTRG.5531; MSTRG.5533; MSTRG.5528; MSTRG.11794; MSTRG.14221; MSTRG.2767; MSTRG.6870; MSTRG.6894*), as illustrated by the Venn diagram in Fig. 9; and amplified except MSTRG.2767; MSTRG.6894; MSTRG.6870; MSTRG.11794. Interestingly, most of the candidates retrieved from 2 or 3 prediction procedures corresponded to cinnamyl alcohol dehydrogenases (CAD)-like. Eleven other MDR candidates were also selected given their genomic location and co-expression profiles (*MSTRG.17429; MSTRG.585; MSTRG.5538; MSTRG.5522; MSTRG.5287; MSTRG.5530; MSTRG.15281; MSTRG.1695; MSTRG.1633; MSTRG.21660; MSTRG.6651*). Lastly, we also extended the search for yohimbane synthase activity to SDR by selecting the three candidates predicted by two distinct approaches but we only amplified *MSTRG.20132*.

For functional analysis, each of these 22 candidates was expressed together with strictosidine-β-D-glucosidase (SGD) from *R. serpentina* in a yeast strain producing strictosidine MIA-CH-A2[52], using a high-copy 2μ plasmid. MIAs produced and secreted

in the yeast medium cultures were then analyzed by UPLC-MS, monitored at the mass-to-charge ratios of yohimbane (*m/z* 355 - positive ionization mode) and heteroyohimbanes (*m/z* 353 - positive ionization mode), and compared to authentic standards (Fig. 9). For heteroyohimbanes, we noted that four candidates (MSTRG.5294, MSTRG.17429, MSTRG.747 and MSTRG.1695) produced significant amounts of a compound co-eluting with tetrahydroalstonine and having a similar fragmentation profile (daughter ions of 222, 210 and 114; Supplementary Fig. S5). Careful examination of the reaction products revealed that MSTRG.5283 also produced minute amounts of tetrahydroalstonine. In addition, we noted that MSTRG.5294, MSTRG.747 and MSTRG.1695 synthesize traces of a compound co-eluting with mayumbine (RT 10.7 min) while MSTRG.17429, MSTRG.747 also synthesized a distinct *m/z* 353 MIA (RT 12.3) whose identity remains unsolved but likely corresponding to a heteroyohimbane based on its fragmentation (Supplementary Fig. S5). Notably, MSTRG.2694 synthesized a dramatically different product profile consisting of a racemate of ajmalicine (RT 10.3 min) and mayumbine. Lastly, we saw that the SDR MSTRG.20132 produces a compound with m/z 353 MIA (RT 9.9 min) that we did not identify.

For yohimbanes, while no biosynthesis was detected for 21 of the candidates at the expected mass of *m/z* 355, we saw that MSTRG.5283 displays a complex enzymatic activity by synthesizing four distinct products. Interestingly, the two main compounds (RT 6.2 min and 7.7 min) co-elute with rauwolscine and yohimbine standards and also display similar fragmentation patterns (Supplementary Fig. S6). One of the two minor products has a similar retention time and fragmentation with corynanthine (RT 7.9 min) but the lack of authentic standard did not allow us to identify the last product (RT 7.0 min). However, fragmentation analysis strongly suggests that this compound also belongs to yohimbanes (Supplementary Fig. S6) and may correspond to alloyohimbine. Overall, a 40/8.5/33.5/18 ratio was observed for rauwolscine/ putative alloyohimbine/yohimbine/corynanthine. Besides its minor tetrahydroalstonine synthase activity, this result thus confirms that *MSTRG.5283* encodes a bona fide yohimbane synthase (YOS) capable of synthesizing multiple yohimbanes as observed with HYS in *C. roseus* for heteroyohimbanes. Surprisingly, we noted that *YOS* is more identical to *THAS1* from *C. roseus* than any other MDR identified in this work while THAS1 has not been reported to produce yohimbanes. In addition, based on their main biosynthetic activity and their identity with the already identify *THAS* from

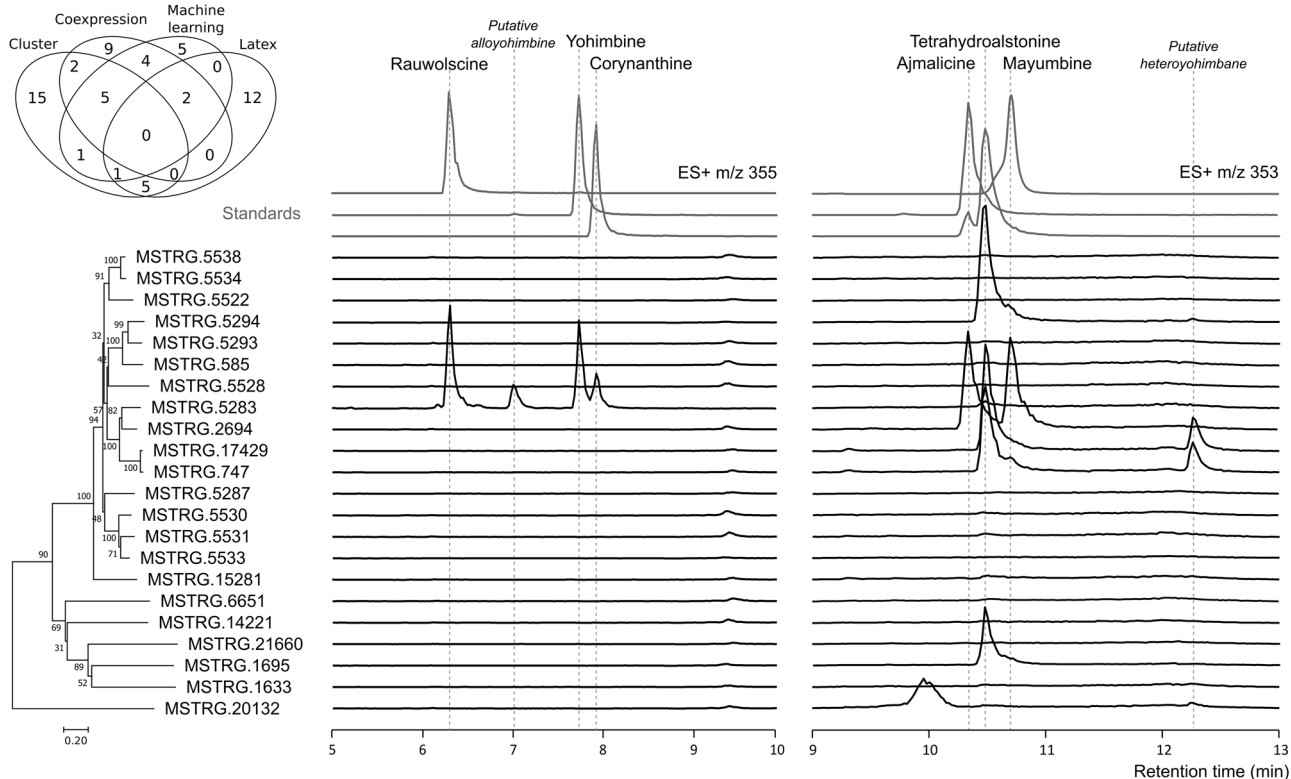

**Fig. 9 Screening of *R. tetraphylla* ADH candidate activity in yeast for yohimbane and heteroyohimbane alkaloids synthesis from strictosidine aglycone.** The Venn diagram regroups the final 61 MIA-related ADH candidates according to their identification methods. 22 ADHs were cloned and expressed in a yeast strain producer of strictosidine aglycones and MIAs accumulated in culture medium were detected by UPLC-MS using positive ionisation mode at m/z 355 (yohimbane) and m/z 353 (heteroyohimbane). The y axes are normalized ion abundance. ADHs are classified using a neighbor-joining phylogenetic tree (100 bootstrap replications). Authentic standard chromatograms are indicated in grey. See Supplementary Fig. S5 and S6 for mass fragmentation of m/z 353 and 355 products respectively.

*C. roseus*, we also named *MSTRG.5294, RtTHAS3* (where Rt denotes *Rauvolfia tetraphylla*); *MSTRG.17429, RtTHAS4A*; *MSTRG.747, RtTHAS4B* (A, B denotes two isoforms) and *MSTRG.1695, THAS5* since no equivalent has been identified to date. Finally, we named *MSTRG.2694*, ajmalicine/mayumbine synthase (*AMS*) in agreement with its reaction products and its high identity with *HYS*.

**Biochemical characterization of candidate MDR suggests distinct yohimbane biosynthetic modalities.** To provide further insights into *YOS* characterization, the corresponding recombinant protein was produced in *E. coli* and assayed in comparison with other enzymes encoded by *RtTHAS3*, previously characterized for its tetrahydroalstonine production, MSTRG.20132 from the SDR family and MSTRG.5538 that did not display any apparent activity in our experimental conditions (Fig. 10, Supplementary Fig. S7). Each of these four enzymes has been assayed together with SGD and strictosidine. In vitro, YOS possesses a similar activity by synthesizing the four m/z 355 MIAs at a similar ratio with rauwolscine as the most abundant product (Fig. 10a). The production of tetrahydroalstonine was also confirmed and we also noted the synthesis of mayumbine in the same proportion. While we could not explain why this last production was not observed in yeast (Fig. 9), the assay with recombinant YOS confirmed the wide yohimbane synthase activity of this enzyme. The ratio of the produced yohimbane/heteroyohimbane (79/21) argue for a main yohimbane synthase activity potentially conferring to YOS a role in the synthesis of these MIAs *in planta*. As a control, we confirmed that MSTRG.5538 remained inactive and was unable to convert deglycosylated strictosidine into downstream MIAs (Supplementary Fig. S7). In these experimental

conditions, we also noted that RtTHAS3 synthesized tetrahydroalstonine with a minor mayumbine synthase activity (Fig. 10a). A low potential heteroyohimbane synthase activity was also monitored in the absence of the cofactor NADPH. Surprisingly, a detailed analysis of the reaction products also revealed a strictly NADPH dependent synthesis of rauwolscine, yohimbine and of the fourth unknown m/z 355 MIA already reported for YOS thus suggesting a potential low yohimbane synthase side activity. Finally, the assay with MSTRG.20132 highlighted the production of a m/z 371 MIA (Fig. 10a) sharing a similar fragmentation pattern with vitrosamine (Supplementary Fig. S8[38]). This result agreed with the high identity of MSTRG.20132 with VAS from *C. roseus*, thus suggesting that the m/z 353 observed in yeast (Fig. 9) corresponds to a dehydration of vitrosamine as previously observed for VAS[38]. While the synthesis of vitrosamine was partially NADPH dependent as reported in *C. roseus*, we also observed the synthesis of a low amount of yohimbine suggesting that MSTRG.20132, hereafter renamed RtVAS, also displays a low NADPH dependent yohimbane synthase side activity (15/85 yohimbine/vitrosamine ratio). All together, these results strongly suggest that different modalities of yohimbane synthesis co-exist. Indeed, while MDRs, mainly YOS and to a lower extent RtTHAS3 can perform the double reduction of the strictosidine aglycon into yohimbanes, SDRs, especially RtVAS, also share a similar side activity. Based on our experimental results and the amount of yohimbanes synthesized by each enzyme (Supplementary Fig. S7), YOS is potentially a major contributor to this synthesis in *R. tetraphylla*. Such a statement is reinforced by the high expression level of YOS in almost all analyzed organs (Supplementary Fig. S9).

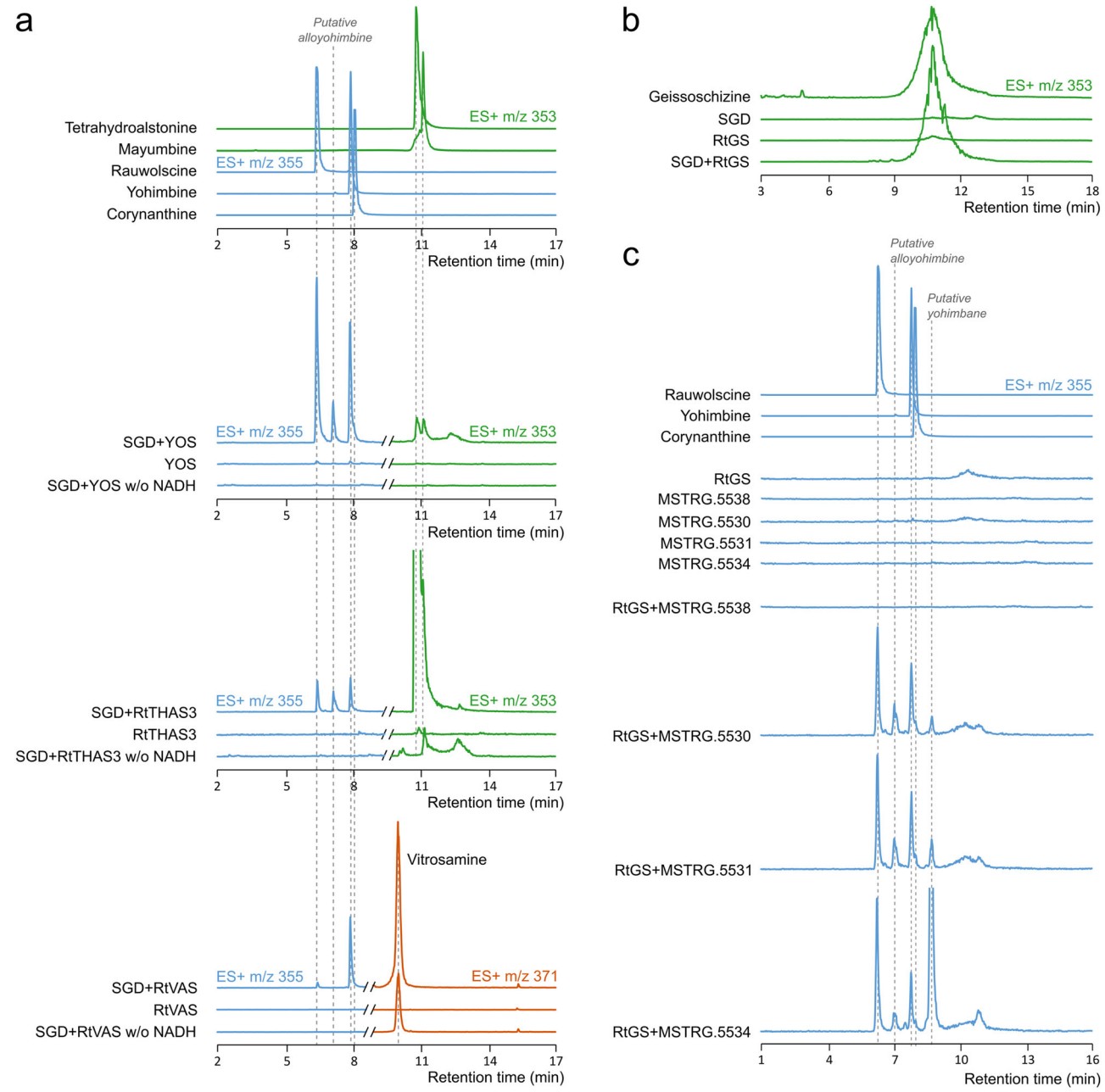

**Fig. 10 In vitro biochemical assays of heteroyohimbane producing ADHs.** Strictosidine was pre-incubated with purified recombinant SGD followed by the addition of recombinant ADHs with or without NADH. Reaction products were analyzed by UPLC-MS and compared to authentic standards. **a** Activities of YOS (MSTRG.5283), RtTHAS3 (MSTRG.5294) and RtVAS (MSTRG.20132) monitored using positive ionisation mode at m/z 355, 353 and 371. **b** Activity of RtGS (MSTRG.5528) monitored using positive ionisation mode at m/z 353. **c** Activities MSTRG.5530, MSTRG.5531, MSTRG.5534 and MSTRG.5538 alone or in combination with RtGS, monitored using positive ionisation mode at m/z 355. Mass fragmentation of products are presented in Supplementary Fig. S5, S6 and S8.

**Associating two distinct reductases for the synthesis of yohimbanes**. Since the synthesis of yohimbanes proceeds through a double reduction of the strictosidine aglycon, we also investigated whether this reaction is more effectively catalyzed by two distinct enzymes. The capacity of GS to reduce strictosidine aglycon, notably 4, 21 dehydrogeissoschizine, prompted us to include GS as a potential partner of a two-enzyme reaction. Given the high identity of *MSTRG.5528* with *GS* from *C. roseus* (Supplementary Table S5), the corresponding recombinant enzyme was first assayed together with *SGD* and strictosidine to investigate its function (Fig. 10a). The formation of a *m/z* 353 MIA coeluting with geissoschizine strongly suggests that MSTR.5528

encodes a *R. tetraphylla* GS (RtGS). Similar assays were next conducted by including RtGS and the same MDRs encoded by the RtGS genomic neighbor we were able to express and purify from *E. coli*, namely *MSTRG.5530, MSTRG.5531, MSTRG.5534, MSTRG.5538*. When assayed alone with SGD and strictosidine, none of these enzymes were able to catalyze the formation of a *m/z* 355 MIA (Fig. 10c). However, when RtGS was added to these assays, we observed the formation of 5 compounds at m/z 355 for MSTRG.5530, MSTRG.5531 and MSTRG.5534, while no production was monitored with MSTRG.5538. Interestingly, three of these compounds co-elute with the authentic standards of rauwolscine, yohimbine and corynanthine and display similar fragmentation

patterns (Supplementary Fig. S6). The fourth compound at RT 7 min probably corresponds to the compound already observed with YOS in yeast and always remained in low proportion. By contrast, the last $m/z$ 355 (RT 9.5 min) showed variable amounts of synthesis and was the most abundant reaction product of the combination of RtGS and MSTRG.5534. While we were not able to firmly identify it, the fragmentation pattern of this compound shares the same daughter ions as yohimbane with additional fragments suggesting that it also belongs to this MIA type (Supplementary Fig. S6). In addition, we observed the presence of a similar compound in different plant extracts thus arguing for a potential physiological relevance of this biosynthetic reaction (Supplementary Fig. S10a, b). In conclusion, our results strongly suggest that a second biosynthetic route of yohimbane may occur *in planta*, which relies on a double reduction process successively catalyzed by RtGS and a downstream MDR including MSTRG.5530, MSTRG.5531 and MSTRG.5534. Comparing the biosynthetic efficiency of the reaction catalyzed by this association with the reaction catalyzed by YOS is not precisely achievable given the experimental differences between the two assays. However, the raw estimation of the yohimbane amounts produced by each reaction is in large favor of YOS, thus arguing again for a major role of this enzyme for the synthesis of yomhimbanes *in planta*. This is only different for the $m/z$ 355 (RT 9.5 min) that was only observed with the two enzyme reactions and notably RtGS and MSTRG.5534, suggesting that this couple of enzymes synthesize this MIA in *R. tetraphylla* roots (Supplementary Fig. S10a, b).

## Discussion

To date, the study of MIA biosynthesis has been mainly conducted in a few but well-known species including *C. roseus* for vindoline and catharanthine, *V. minor* for vincamine, *C. acuminata* for camptothecin or *R. serpentina* for ajmaline. For the first three species, high quality genomes have been published and have guided the elucidation of specific steps MIA biosynthesis as elegantly conducted in *C. roseus*[21]. In the present study, we have first described the sequencing of *R. tetraphylla*, a related *R. serpentina* species, well known for MIA synthesis. This genome is characterized by a very high contiguity and completeness. We annotated a total of 23,228 genes which is comparable to other published Apocynaceae genomes. Gene family identification, phylogenetic relationships and gene family contraction and expansion analysis showed an enrichment in genes associated with specialized metabolism, notably indole metabolism in *R. tetraphylla*. Regarding MIA synthesis, we also noted the presence of several potential MIA clusters somehow similar to what has been observed in *C. roseus* thus arguing for a conservation of the genomic organization of MIA biosynthetic genes. Interestingly, and similarly to previous observations in *C. roseus*[21,22], we noted a putative local duplication of an ADH rich cluster in *R. tetraphylla* (Supplementary Fig. S3). Such duplication could explain the enrichment in genes associated with indole metabolism in *R. tetraphylla*. These duplications could also be the source of enzyme subfunctionalization leading to the diversity and molecular variability found in closely related AIM-producing species.

Given the high abundance of rauwolscine in *R. tetraphylla*, we engaged in the identification of the genes/enzymes involved in the synthesis of yohimbanes since this MIA type has not been documented at the biosynthetic level to date. Using this new genomic resource, the associated transcriptomic analyses and a targeted proteomic studies, we combined the study of the gene expression correlation to the prediction MIA biosynthetic genes guided by machine learning or based on their genomic localization to unravel the actor of yohimbane synthesis. Based on the mechanism of heteroyohimbane and strychnos formation, we focused on proteins

from different ADH families, notably MDR and SDR that have already been associated to MIA synthesis[28,36–38]. By combining activity testing in yeast to biochemical assays using recombinant enzymes, we first identified four enzymes displaying a high tetrahydroalstonine synthase activity (RtTHAS3, RtTHAS4A, RtTHAS4B, THAS5) and noticed that these proteins also synthesize at lower amounts mayumbine and another unknown MIA (RT 12.3) (Figs. 9, 10). Interestingly, in our experimental conditions, AMS produces a racemic mixture of ajmalicine and mayumbine in agreement with its high identity with *HYS* from *C. roseus*[37] and the presence of these MIA *in planta* (Figs. 5b, 9). However, no production of tetrahydroalstonine was monitored suggesting a slightly different catalytic mechanism compared to *HYS*. Overall, the efficient biosynthetic activity of AMS combined to its very high expression level in leaf argue for a main role of AMS in the synthesis of ajmalicine and mayumbine in the aerial parts of the plant. By contrast, RtTHAS3, RtTHAS4A and RtTHAS4B are probably the main contributors to the synthesis of tetrahydroalstonine and its abundant derivatives including reserpilline and isoreserpilline in plant leaves, based on their expression profiles (Supplementary Fig. S9). This finding thus reinforces the role of MDR and especially CAD-like in MIA biosynthesis. However, we also showed that a non-CAD MDR (THAS5) can synthesize heteroyohimbanes suggesting that distinct evolutionary processes led to the formation of this biosynthetic activity.

Secondly, using the same screening approaches, we identified YOS (MSTRG.5283) as a multi-functional yohimbane synthase. While comparison of its biosynthetic products to authentic standards confirms the synthesis of rauwolscine, yohimbine and corynanthine, MS/MS analysis also suggests that the four m/z 355 MIA correspond to the fourth isomer named allo-yohimbine (Figs. 1, 10a). Interestingly, we noted that YOS preferentially synthesizes rauwolscine and yohimbine and to a lower extent their respective stereoisomer, in addition to a very minor tetrahydroalstonine synthase activity. As such, *YOS* can be considered as a counterpart of *HYS* for yohimbanes even if closer to *THAS1* at the amino-acid sequence level (Supplementary Table S5). This identification of *YOS* thus corresponds to the first description of the yohimbane biosynthetic modalities and its high identity with enzymes synthesizing heteroyohimbanes suggest that they have evolved from common ancestor.

Surprisingly, a careful examination of the reaction products generated in vitro using recombinant enzymes revealed that RtTHAS3 (MSTRG.5294) and RtVAS (MSTRG.20132) also produce yohimbanes in our experimental conditions, besides their main and respective tetrahydroalstonine and vitrosamine synthase activity (Fig. 10a). This production is of a similar order of magnitude for both enzymes but remains low compared to *YOS*, (Supplementary Fig. S7). Such a result has thus multiple implications by suggesting that (i) yohimbane synthesis can be widespread in MDRs, (ii) MIA biosynthetic enzymes can be both heteroyohimbane and yohimbane synthases in agreement with a possible common ancestor and (iii) SDRs, at least RtVAS, can also synthesize yohimbane namely yohimbine. From a mechanistic point of view, it is highly likely that each of these three enzymes proceed through a similar mechanism relying on a two step-reduction of 4,21 dehydrogeissoschizine, as already suggested for GS and the formation of (16*R*)-*Z*-isositsirikine and (16*R*)-*E*-isositsirikine alone or in combination with redox 2[28,53]. Indeed, 4,21 dehydrogeissoschizine has been proposed to be in equilibrium with a dienamine form[54]. The formation of yohimbane would thus require the reduction of the C16-C17 leading to ring closure and followed by the iminium reduction that appears to be a common feature of MDRs (Supplementary Fig. S11)[27]. Such a scenario thus suggests a competition between *GS* and *YOS* for 4, 21 dehydrogeissoschizine (or dienamine) to direct MIA synthesis towards sarpagan or

yohimbane MIA types. THAS, HYS and the related AMS reduce cathenamine, which results from the spontaneous cyclisation of 4,21 dehydrogeissoschizine[36,37]. We hypothesize that YOS traps 4, 21 dehydrogeissoschizine before spontaneous cyclisation to cathenamine occurs, as also true for GS.

Based on the localization of SGD, the release of strictosidine aglycone isomers including 4,21 dehydrogeissoschizine occurs in the cell nucleus[25,26]. While some THAS displays class V nuclear localization sequences (NLS)[36,37], neither YOS nor GS possess any observable localization sequences to efficiently target the proteins to this subcellular compartment. In this context, it is obvious that this spatial distribution favors the synthesis of tetrahydroalstonine and derivatives (reserpilline and isoreserpilline) that are highly accumulated in the aerial parts of R. tetraphylla. By contrast, in roots where ajmaline is the most abundant MIA, with low amounts of tetrahydroalstonine derivatives, huge expression of GS coupled to lower expression of THAS would guarantee access to 4,21 dehydrogeissoschizine (Supplementary Fig. S9). Similarly, differences in gene expression level and the abundance of the corresponding proteins may also provide access to 4,21 dehydrogeissoschizine/dienamine to enzymes synthesizing yohimbanes. For instance, we noted the presence of a very high amount of rauwolscine in leaf latex (Fig. 7b) that correlates with the presence of YOS. It thus suggests that YOS could be preferentially expressed in the leaf laticifers to ensure the synthesis of yohimbanes in this cell type[55]. Interestingly, the presence of enzymes involved in alkaloid biosynthesis has been well-documented in latex for benzylisoquinoline alkaloids[56] but never for MIAs to date.

Although yohimbanes can be synthesized from strictosidine aglycone using only YOS, our in vitro assays also revealed that these MIAs can be also synthesized through a two-enzyme system involving RtGS and the proteins encoded by genome neighbors including GS close genomic neighbors (Figs. 8, 10). Although the results obtained by the different biochemical assays are difficult to compare, the amount of yohimbanes produced by two enzymes appears to be in the same order of magnitude as the ones generated by RtTHAS3 or RtVAS alone. While the catalytic mechanism could also rely on similar reductions, here catalyzed by each partner, the synthesis of an additional MIA (RT 9.5 min), which is the main reaction product of the RtGS and MSTRG.5534 couple, also suggests a higher plasticity of this biosynthesis mode and specific in planta role for the synthesis of this compound accumulated in roots (Supplementary Fig. S10a, b). Interestingly, we observed that gene encoding each of the four enzymes RtGS, MSTRG.5530, MSTRG.5531, MSTRG.5534 are always co-expressed in the different R. tetraphylla organs thus arguing in favor of the in vivo existence of this two-enzyme biosynthesis.

Overall, the respective role of each enzyme (YOS, RtTHAS3 and VAS) to the global synthesis of yohimbanes in the whole plant is an open matter of debate. Based on their catalytic efficiency measure in vitro and their gene expression level (Supplementary Fig. S7, S9), YOS is likely to be a major contributor to yohimbane production, while VAS acts specifically for a high and specific synthesis of yohimbine. Conversely and despite a very high expression level, the lack of NLS in RtTHAS3 does not allow an efficient targeting of this enzyme to the nucleus and the capture of 4,21 dehydrogeissoschizine/dienamine. In doing so, it would rather limit the role of RtTHAS3 in yohimbane synthesis.

From an evolutionary point of view, the multiplicity of the yohimbane synthesis modes raises questions. Of course, we cannot exclude the existence of these different modes simply results from substrate promiscuity and side activity of the enzymes tested as already observed for tabersonine 3-oxygenase and vindorosine synthesis[19,57]. By contrast, the co-occurrence of single or two-enzyme mechanisms for yohimbane synthesis could also be considered as a nice example of the diversification of the biosynthetic mechanisms that the plant evolves to generate this MIA type. Such a diversification may have arisen from the local duplication of ADHs leading to the emergence of genes encoding THAS, GS but also YOS and VAS in a close genomic environment (Fig. 8). While synteny is conserved with C. roseus, we noted a higher degree of ADH multiplication for R. tetraphylla in this specific region (Supplementary Fig. S3). This local duplication seems to be a common feature of MIA evolution and has been already reported for tabersonine 16-hydroxylases in the synthesis of vindoline[19,58] and several BAHD acylating MIA including tabersonine derivatives[21,59]. Obviously, the future characterization of ADHs from similar genomic regions in other MIA producing plants will potentially wider MIA synthesis characterization, ultimately leading to the identification of new MIA biosynthesis actors.

In conclusion, by sequencing the genome of R. tetraphylla and combining distinct MIA candidate gene prediction approaches, we identified several enzymes catalyzing the synthesis of yohimbanes. While the use of a single prediction approach may result in the identification of interesting candidates such as YOS in the latex proteome, integration of several strategies including ML based approach, co-expression analysis and gene cluster screening first reinforced the aforementioned enzyme identification but also widened discovery to unexpected biosynthetic mechanism such as yohimbane synthesis through a double enzyme process. Such a result thus adds another level of understanding to the MIA synthesis resulting from strictosidine aglycone reductions, besides the already described strychnos and heteroyohimbane MIAs. While several mechanisms of yohimbane synthesis may co-occur in planta, the high activity of YOS in yeast potentially paves the way for the synthesis of rauwolscine and other yohimbane in yeast as already described for other highly valuable MIAs and their halogenated derivatives[52,60]. This is especially required to exploit the selective affinity of rauwolscine and corynanthine for α2-adrenergic and α1-adrenergic receptors, respectively.

## Materials And methods

**Sample collection, DNA extraction and sequencing**. *Rauvolfia tetraphylla* seeds (Konstanz botanical garden, Germany) were germinated and planted in individual pots. Seedlings were allowed to grow at 28 °C under a 16 h light/8 h dark cycle. DNA was extracted from young leaves of two-month-old plants using Qiagen Plant DNeasy kit (Qiagen, Hilden, Germany) following the manufacturer's instructions. DNAseq library was performed by Future Genomics Technologies (Leiden, The Netherlands) using Nextera Flex kit (Illumina, San Diego, USA) for Illumina sequencing and ONT 1D ligation sequencing kit (Oxford Nanopore Technologies Ltd, Oxford, United-Kingdom) for Nanopore sequencing. Illumina libraries were sequenced in paired-end mode (2 × 150 bp) using Illumina NovaSeq 6000 technology. ONT libraries were sequenced on Nanopore PromethION flowcell (Oxford Nanopore Technologies Ltd, Oxford, United-Kingdom) with the guppy version 3.0.3 high-accuracy basecaller.

**De novo genome assembly**. The *R. tetraphylla* genome assembly was performed by Future Genomics Technologies (Leiden, The Netherlands). ONT reads were first assembled using Flye assembler (v.2.5[61]). Contig were twice corrected with ONT reads using Flye (v.2.5) and subsequently polished twice with Illumina reads using pilon (v.1.23[62]). Redundant contigs were removed using purge_haplotigs (v.1.1.0) followed by a last round of polishing with Illumina reads using pilon.

**Biotic interaction between *R. tetraphylla* leaves and *Manduca sexta***. *R. tetraphylla* seeds (Konstanz botanical garden, Germany) were germinated and planted in individual pots. Seedlings were

allowed to grow at 28 °C under a 16 h light/8 h dark cycle for 4 months or 10 weeks in case of folivory studies. Rearing of *M. sexta* larvae, as well as feedings were performed as described in ref. [5]. For the experimental setup described in this study, the leaf areas around the *M. sexta* fed regions (+/− 5–10 mm) were collected by cutting with a scalpel. Samples were collected before *M. sexta* exposure (Leaf - 0 h), and then at 24 h (Bitten leaf - 24 h), including control Leaf (24 h) and 48 h (Bitten Leaf - 48 h), including control (Leaf - 48 h) after the beginning of the experiment. Samples were collected in quadruplicate.

**RNA extraction and sequencing.** RNA was extracted from 94 different *R. tetraphylla* samples, obtained from 94 different tissue types and experimental conditions including: small tetrad leaves, big tetrad leaves, stems, flowers, berries, young leaves, roots, first leaf pairs, leaves and roots grown under high and low light conditions and leaves fed on by *M. sexta* (Supplementary Data S5). Samples were flash frozen in liquid nitrogen and RNA was extracted using the Macherey-Nagel NucleoSpin® RNA plant and fungi kit (Düren, Germany) as per manufacturer's instructions. The RNA library construction and sequencing were performed at FGTech using Illumina NovaSeq 6000 technology. Sequencing data has been deposited under the Bioproject accession number: PRJNA771251.

**Gene model prediction and functional annotation.** RNA-seq reads were processed using FastP (v0.20[63]) with default settings. The resulting reads were then aligned to the *R. tetraphylla* reference genome using HiSAT2 (2.2.1[64]) for each of the 94 individual samples. Each individual alignment was then assembled into individual transcriptomes using StringTie (v2.1.7[65]). The resulting 94 individual transcriptomes were then merged into a non-redundant set of 56,389 representative transcripts using stringtie --merge. Functional annotation of the consensus transcriptome was achieved with the Trinotate pipeline (v3.0.1[66],), which integrates results from blastp and blastx searches of TransDecoder (v5.5.0[67],) predicted ORFs against the Uniprot database, and hmmscan (v3.1b2[68]) against the PFAM database (https://pfam.xfam.org/).

**Assembly completeness assessment.** Assembly quality assessment was performed combining the stat program from BBMap tool (v.38.94[69]), MerQury (v. 1.3[70]) and LTR assembly index from LTR_retriever (v2.9.6[71]). Benchmarking Universal Single-Copy Orthologs (BUSCO v.5.2.2[72]) with default settings using a plant-specific database of 2,326 single copy orthologs (eudicots_odb10) was used to assess assembly and gene models completeness. The agat_sp_statistics from the AGAT package (v.0.8.0[73]) allowed us to get gene models statistics.

**Transposable elements prediction and annotation.** To identify and annotate transposable elements (TE), Extensive de novo TE annotator (EDTA v.1.9.5[74]) was used using sensitive and evaluate options.

**Whole-genome duplication analysis.** The DupPipe pipeline[75] was used to infer whole genome duplication (WGD) events using transcript sequences of *R. tetraphylla*, *V. thouarsii*[18], *V. minor*[17], *C. roseus*[20], *Arabidopsis thaliana*[76], *Mytragyna speciosa*[15], *Solanum lycopersicum*[77], *C. acuminata*[13], *Calitropis gigantea*[48], *G. sempervirens*[12], and *O. pumila*[14]. To identify duplicated gene pairs (40% sequence similarity over 300 bp), discontiguous MegaBLAST[78,79] was used on each dataset. Open reading frame of each gene pair was inferred from the NCBI's plant RefSeq protein database (May 21, 2021) using BLASTx (v.2.6.0-1[80]) retaining the best hit sequence only (sequence similarity

threshold: 30% over 150 amino acids). GeneWise[81] subsequently performed DNA sequence alignment against the best hit homologous protein and its translation. MUSCLE (v.3.6[82]) performed amino acid sequence alignment for each gene pair which further guided nucleic acid alignment using RevTrans (v.1.4[83]). Finally, Codeml's F3x4 model from PAML package (v.4.9[84]) was used to calculate substitutions per synonymous site (Ks) and thus determine divergence times between gene pairs.

**Orthology analysis and phylogenetic tree reconstruction.** To build gene families, protein sequences of at least 30 amino acids from *R. tetraphylla* were compared with eleven other species including *V. thouarsii*[18], *V. minor*[17], *C. roseus*[20], *A. thaliana*[76], *M. speciosa*[15], *S. lycopersicum*[77], *C. acuminata*[13], *C. gigantea*[48], *G. sempervirens*[12], and *O. pumila*[14]. For each species, the longest representative protein was selected in each CD-HIT (v.4.7[85]) cluster. The resulting sequences were used as input for Ortho-Finder (v.2.5.4[86]) using the following parameters: -S diamond -M msa -A muscle -T raxml-ng. A maximum-likelihood phylogenetic tree was built from 645 single-copy orthogroups. Cafe5 (v.4.2.1[87]) was used to determine orthogroup loss and expansion across the phylogenetic tree.

GO term enrichment on expanded orthogroups in *R. tetraphylla* was performed by comparing the relative occurrence of a GO term into the increased orthogroups gene list to its relative occurrence in the genome using a Fisher's exact test (2-sided) with the R function topGO (v.2.44.0[88]). Cut-off criterion was set to a Benjamini-Hochberg adjusted *P*-value of 0.05. Enriched terms graphs were performed using ggplot2 (v.3.3.5[89]).

**Genome-wide synteny analysis.** Minimap2 (v.2.24[90]) was used to align genomes of *R. tetraphylla* and *C. roseus* v.2.1[20] with the following options: -cx asm20 –cs. D-Genies[91] was used to visualize the obtained paf file selecting hits with minimum 80% identity and sorting contigs by size.

**Proteome analysis of *R. tetraphylla* latex and leaves.** Leaves from three-months-old plants were manually cut to allow the latex to pour out. The latex was collected with a pipette tip and immediately dissolved in the protein precipitation buffer according to[92]. Briefly, around 100 µL of leaf latex was diluted in 100 µL of SDS extraction buffer (2% SDS, 60 mM DTT, 40 mM Tris-HCl pH 6.8). Once vortexed, the samples were heated at 100° for 5 min, proteins were precipitated by adding half the volume of trichloroacetic acid/acetone 1:1 with 60 mM DTT and incubating at −20° for 1 h 30 min. Samples were centrifuged at 16000 g for 20 min at 4° and the dried pellet was washed three times with ice-cold 80% acetone supplemented with 60 mM DTT. The dried Speed-vac pellet was resuspended in 50 µL of Laemmli buffer. Proteins from the corresponding leaves were also extracted after leaf grinding in the described extraction buffer. Proteins were separated on the denaturing 10% acrylamide gel and analyzed with Colloidal Comassie Blue (CCB) under sterile conditions[93]. 1-cm large strips were cut off the gel and sent to the PAPPSO Platform (INRAE Moulon) for in gel digestion, LC MS/MS analysis and peptide identifications.

**Protein in gel digestion.** Gel pieces were washed thrice by successive separate baths of 10% acetic acid, 40% ethanol and 25 mM ammonium bicarbonate and ACN. Proteins were reduced for 30 min with 10 mM dithiothreitol, 25 mM ammonium bicarbonate at 56 °C and alkylated in the dark with 55 mM iodoacetamide, 25 mM ammonium bicarbonate for 45 min at room temperature. The gel pieces were washed by successive separate baths of 0.1 M ammonium bicarbonate and ACN. Digestion was

subsequently performed overnight at 37 °C with 125 ng of modified trypsin (Promega) dissolved in 50 mM ammonium bicarbonate. The peptides were extracted successively with 0.5% trifluoroacetic acid (TFA) and 50% ACN and then with ACN. Peptide extracts were dried in a vacuum centrifuge.

**LC-MS/MS analysis of peptides**. Peptide samples were solubilized in a buffer of 2% CH3CN, 0.1% formic acid. Liquid chromatography was performed on a NanoLC Ultra system (Eksigent, Dublin, CA, USA). Samples were loaded at 7.5 µl min$^{-1}$ on a C18 precolumn (5 µm, 100 µm i.d. ×2 cm length; NanoSeparations) connected to a separating BIOSPHERE C18 column (3 µm, 75 µm i.d. ×300 mm length; NanoSeparations). Solvent A was 0.1% formic acid in water and solvent B was 0.1% formic acid in CH3CN. Peptide separation was achieved using a linear gradient from 5 to 35% solvent B for 28 min at 300 nl min$^{-1}$. Including the regeneration and the equilibration steps, a single run took 45 min. Eluted peptides were analyzed with a Q Exactive™ Plus mass spectrometer (Thermo Fisher Scientific) using a nanoelectrospray interface. Ionization was performed with a 1.3 kV spray voltage applied to an uncoated capillary probe (10 µm i.d., New Objective). The Xcalibur interface was used to monitor data-dependent acquisition of peptide ions. This included a full MS scan covering a mass-to-charge ratio (m/z) of 350 to 1400 with a resolution of 70,000 and an MS/MS step (normalized collision energy, 27%; resolution, 17,500). The MS/MS step was reiterated for the eight major ions detected during the full MS scan. Dynamic exclusion was set to 50 s. Only doubly and triply charged precursor ions were subjected to MS/MS fragmentation. The mass spectrometry proteomics data have been deposited to the ProteomeXchange Consortium via the PRIDE partner repository with the dataset identifier PXD046315.

**Identification of peptides**. A database searches were performed using X!Tandem Alanine (Release 2017.2.1.4) (http://www.thegpm.org/TANDEM). Enzymatic cleavage was described to be due to trypsin digestion with one possible misscleavage. Cys carboxyamidomethylation was set as static modification whereas Met oxidation, N-ter deamidation and N-ter acetylation were set as variable modifications. Identifications were performed using a user-supplied database and an internal database of standard contaminants (trypsin, keratins, BSA). Identified proteins were filtered and grouped using X!TandemPipeline v0.2.38 (http://pappso.inrae.fr/bioinfo/i2masschroq/)[94]. Data filtering was achieved according to a peptide E value smaller than 0.01 with a minimum of 2 peptides to identify a protein.

**Metabolite profiling of R. tetraphylla**. Metabolites were extracted from 94 different *R. tetraphylla* samples, obtained from 94 different tissue types and experimental conditions, described above in sections *Biotic interaction between R. tetraphylla leaves* and *RNA extraction and sequencing* (Supplementary Data S5). Samples were frozen in liquid nitrogen, then freeze-dried, ground into powder, and sonicated with methanol 0.1% of formic acid to extract metabolites. For *R. tetraphylla* latex, around 200 µL of latex from freshly cut leaves were diluted in 200 µL of PBS buffer and the alkaloids were extracted by adding one volume of methanol 0.1% of formic acid. After centrifugations, extracts were diluted in water Milli-Q 0.1% of formic acid and injected on a UPLC system (Acquity, Waters) coupled to a single quadrupole mass spectrometer with an 18-min linear gradient from 10 to 40% acetonitrile (containing 0.1% formic acid). Separation was performed using a Waters Acquity HSS T3 C18 column (150 mm * 2.1 mm, 1.8 µm) with a flow rate of 0.4 ml.min$^{-1}$ at 55 °C, with an injection volume of 5 µL. Mass spectrometry detection was performed using an SQD mass spectrometer

(SQD2, Waters), with capillary and sample cone voltages of 3000 V and 30 V respectively, and cone and desolvation gas flow rates of 60 L.h$^{-1}$ and 800 L.h$^{-1}$. The selected mode of ion monitoring was employed in positive mode for the following compounds: ajmalicine (m/z 353), RT = 10.4; ajmaline (m/z 327), RT = 7.4, alstonine (m/z 349), RT = 11.7; corynanthine (m/z 355), RT = 7.9; isoreserpiline (m/z 413), RT = 9.0; loganic acid (m/z 377), RT = 2.1; rauwolscine (m/z 355), RT = 6.3 ; reserpiline (m/z 413), RT = 10.1; serpentine (m/z 349), RT = 11.6; strictosidine (m/z 531), RT = 9.0; tetrahydroalstonine (m/z 353), RT = 10.6 ; tryptamine (m/z 161), RT = 3.0 ; vinorine (m/z 335), RT = 8.6; yohimbine (m/z 355), RT = 7.7.

**Expression atlas and differential gene expression analysis**. Abundance estimates were established as transcripts per million (TPM) by pseudo-aligning and counting the RNA-seq reads for each of the 94 samples to the transcriptome with Salmon (v0.14.1[95]) with the -biasCorrect and -vbo flags. The resulting expression profiles parsed into a combined expression matrix.

Differentially expressed genes (DEGs) were determined for *M. sexta* exposed *R. tetraphylla* leaves by fitting a quasi-likelihood negative binomial generalized log-linear model to the expression matrix with the edgeR package (v3.28.1[96]). For the following contrasts: 24 h vs 24 h control (Ms24h/Ctrl24h) and 48 h vs 48 h control (Ms48h/Ctrl48h), transcript wise exact tests were performed and transcripts were considered to be differentially expressed if the *p*-value was below 0.05, and transcripts with a log fold change above 0 were considered to be upregulated.

**Co-expression network analysis**. A matrix containing both the gene and the metabolite accumulation profiles was created using the paired samples. To ensure a similar distribution of the values in samples, metabolite data were first converted into parts per million (the accumulation value divided by the sum of values for the same sample, multiplied by one million). Transcripts were retained if they had a TPM value above 10 in at least more than 6 samples. A Pairwise Pearson's correlation coefficient was calculated for each gene-gene, gene-metabolite and metabolite-metabolite pair in the transcriptome and ranked as highest reciprocal ranks (HRR) as previously described[97,98]. Highest ranking co-expressed genes were captured for each MIA-like homolog with HRR < 30. Relationships between co-expressed genes were visualized with the igraph package in R (v1.2.6, igraph). Known MIA genes from other species were used as bait to query the global co-expression network. Co-expressed neighbors were collected to identify potentially co-expressed new MIA genes.

**Deep learning classification**. To predict the MIA-related genes from our gene expression atlas with ANN, we referred to our recently described protocol[99]. The input dataset was the same than the one used to construct co-expression network. The H2O library (v3.36) in R (v4.2) was used to train a feedforward neural network with backpropagation. To define the true positive events, a number of genes were labeled as MIA as described above. For the true negative events (non-MIA related genes), we used genes predicted as conserved orthologs from the BUSCO evaluation made to assess assembly completeness. Because these genes are strongly conserved across very diverse plant families, they are not expected to belong to the MIA pathway. We finally labeled 75 genes as MIA-related and 1908 as non-MIA related. This set was split into a training and a validation dataset following a 70/30 partition with seed 666. Once a network (seed 666) was trained, we looked at the number of MIA predicted genes using the full dataset (with data unseen by the network). The logloss was used as a stopping metric. A hyperparameter search revealed that a simple architecture containing 1

input layer (57 samples), 2 hidden layers (with 40 and 20 neurons, respectively) and an output layer containing 2 values (MIA vs non-MIA) provided the most relevant predictions. We added dropout ratios (0.1 in the input and 0.5 in each hidden) to improve the model generalization. The number of epochs was selected by visually inspecting the logloss progression during iterations. A value of 1500 was found to be the most appropriate to avoid overfitting. Calculations were done on a classical computer device (8 CPU, RAM 16 Gb).

**Identification of physically co-localized MIA-like gene regions**. A personal script (accessible at https://doi.org/10.6084/m9.figshare.20749096.v1) was used to identify regions of physically co-localized biosynthetic genes associated with MIA-annotated genes[17]. As input the script depends on (i) a list of blast hits vs known MIA genes, the (ii) resulting genome gtf file from stringtie --merge containing the structural definition of transcripts assembled from read alignments against the genome, and (iii) uniprot search results. In brief, the script first anchors the putative MIA orthologs blastn result and the pfam annotations onto the *R. tetraphylla* genomes gtf file. Next, 100,000 bp regions are scanned on either sides of the MIA annotation to search for any genes with the following pfam accessions annotated: PF03171 (2OG-Fe(II) oxygenase superfamily), PF14226 (non-haem dioxygenase in morphine synthesis N-terminal), PF00891 (*O*-methyltransferase domain), PF08240 (Alcohol dehydrogenase GroES-like domain), PF00067 (Cytochrome P450), PF08031 (Berberine and berberine-like), and PF00201 (UDP-glucosyl transferase). Regions with more than one gene of interest (MIA + pfam accession of interest) were recorded as a cluster of interest.

Microsyntenic region with the two ADH rich clusters of R. tetraphylla (contig 414) was identified in *C. roseus* genome v.2.1 using BLASTN (v.2.6.0-1[80]) with the following parameters: blastn -outfmt 6 -task blastn -perc_identity 70 -evalue 1e$^{-10}$. Hits with an E-value of at least 1e$^{-6}$ and alignment length of at least 1 kb were visualized using the R genoPlotR library (v0.8.11[100]).

**Chemicals**. Chemicals used in this study include strictosidine (Phytoconsult), rauwolscine (Extrasynthese), yohimbine (Sigma-aldrich), corynanthine (Biosynth), tetrahydroalstonine (Extrasynthese), ajmalicine (Fluka) and mayumbine (The BioTek).

**Yeast strains**. The ADH candidates were ordered from Integrated DNA technologies with a set of universal overhangs at both the 3' and 5' end to facilitate easy amplification with a single set of universal primers (BJL167 & BJL171; Supplementary Table S6). Each sequence was cloned into a 2 µ plasmid backbone by use of USER-cloning, with a TRP5 selection marker, and a bi-directional promoter system allowing multicopy co-expression of RseSGD and the ADH's of interest. The resulting plasmids, pBJL66–77 and pBJL118–149 (Supplementary Table S7) were propagated in DH5a competent cells and all plasmids were verified by sanger sequencing prior to transformation. Each plasmid was transformed into MIA-CH-A2[52] using standard LiAc/Heatshock methods according to Gietz and Schiestl[101].

**Culture conditions**. To test the yeast strains expressing ADH candidates for production, six colonies of each strain were inoculated in 150 µL synthetic complete medium lacking tryptophan (SC-Trp) and incubated overnight at 30 °C and 300 rpm in 96-well microtiter plates. After 16 h, 5 µL of each culture was transferred into 0.295 mL of SC-Trp supplemented with 0.25 mM secologanin and 1 mM tryptamine in deep well plates. The plates were then incubated for 96 h at 30 °C and 300 rpm. At the end of the culture, yeast were removed from the medium by ultrafiltration through an

acroprep filter plate at 5000 g for 5 min. Supernatant was mixed with 9 volumes of methanol and 5 µl was analyzed on UPLC-MS as described above in *Metabolite profiling of R. tetraphylla*.

**Production of recombinant ADH and in vitro assays**. Coding sequences of candidates ADH were amplified by polymerase chain reaction (PCR) using their respective primers (Supplementary Table S6) and were subsequently cloned in the pRSET-A (Invitrogen) plasmid. *E. coli* BL21 (DE3) were then transformed using the recombinant plasmid. Transformed bacterial strains were cultivated until exponential growth (Abs=0.6) and protein expression was induced by adding 0.5 mM IPTG for 3 h at 30 °C. *E. coli* were subsequently lysed using a 50 mM potassium phosphate buffer pH 7.5 in presence of 300 mM NaCl, 10 mg/mL lysozyme and Sigma EDTA-free protease inhibitor cocktail. After incubation at room temperature for 20 min, 20 u of DNaseI (Invitrogen) were added and lysate was incubated further for 10 min at room temperature. Sonication of the samples was then done on ice, with an amplitude of 40% and duty cycle of 20 s/40 s using a SONOPULS UW100 (Bandelin) for 3 min. Sonication was repeated twice. Bacterial lysate was clarified by centrifugation at 12,000 g, 4 °C for 20 min and recombinant proteins were purified on Co$^{2+}$ matrix following the manufacturer's protocol (TALON metal affinity resin, TaKaRa). Eluate was desalted using PD10 columns (GE Healthcare) in a 50 mM potassium phosphate pH 7.5 buffer with 100 mM NaCl and 10% glycerol. They were subsequently concentrated when necessary using centricon (30 kDA cut-off, Merck Millipore). Protein concentration was measured according to[102]. Purity of the protein was confirmed on a denaturing 10% acrylamide gel and analyzed with CCB[93].

For in vitro assays, strictosidine aglycones used as ADH substrates were obtained by incubating 50 µM of strictosidine and 0,5 µg of recombinant SGD in 10 µL of a 80 mM citrate buffer pH 6 for 20 min at 30 °C. Afterward, 1 mM NADH, 50 mM potassium-phosphate pH 7,5 buffer and 1–3 µg of purified ADH were added in a final volume of 100 µL. Reaction was stopped with one volume of methanol after 0 or 1 h of incubation at 32 °C. To remove the proteins, samples were centrifuged for 15 min at 12,000 rpm. The supernatant recovered and 5 µL were used for UPLC-MS analysis as described above in *Metabolite profiling* of *R. tetraphylla*. SGD control was performed by adding 20 µL of HCl 1 M prior to the first incubation.

**Statistics and reproducibility**. General information on how statistical analyses were conducted, including software and packages versions and number of replicates (3–7), are described in the relevant sections. Statistical analysis and graphical representation was performed in R[103] using several packages including ggplot2 (v.3.3.5[89]), topGO (v.2.44.0[88]), edgeR (v3.28.1[96]), igraph (v1.2.6) and H2O library (v3.36).

**Gene accession numbers**. *RtTHAS5* (OR514622); *RtTHAS4A* (OR514623); *RtTHAS4B* (OR514624); *RtTHAS3* (OR514625); *RtYOS* (OR514626); *RtVAS* (OR514627); *RtAMS* (OR514628); *RtGS* (OR514629); *MSTRG.5534* (OR514630); *MSTRG.5531* (OR514631); *MSTRG.5530* (OR514632).

**Reporting summary**. Further information on research design is available in the Nature Portfolio Reporting Summary linked to this article.

## Data availability
Raw DNA-seq, RNA-seq and the genome assembly have been deposited in the NCBI database under the BioProject accession number: PRJNA771251 (https://www.ncbi.nlm.

nih.gov/bioproject/PRJNA771251). Raw proteomics data are available via
ProteomeXchange with identifier PXD046315. The genome annotation, coding
sequences, protein sequences, transcript sequences and expression matrix are available
on the figshare: https://doi.org/10.6084/m9.figshare.21679601. Source data for Figs. 2b,
2c, 4b, c, 5b–e, 6c, 7b, d, 9 and 10 are available on Supplementary Table S1,
Supplementary Data S3, Supplementary Data S4, Supplementary Data S6, Supplementary
Table S4, Supplementary Data S6, Supplementary Data S8, Supplementary Data S13 and
Supplementary Data S14, respectively.

## Code availability

H2O and HRR codes are available at https://github.com/EA2106-Universite-Francois-Rabelais.

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

## Acknowledgements

This work was supported by EU Horizon 2020 research and innovation program [MIAMi project-Grant agreement N°814645]; BII Foundation (grant nos. NNF19OC0055591 and NNF20SA0067054); ARD CVL Biopharmaceutical program of the Région Centre-Val de Loire [ETOPOCentre project]; ANR [project MIACYC – ANR-20-CE43-0010]; and APR-IR of the Région Centre-Val de Loire [BioSynNAC project]. The authors benefited from the use of the cluster at the Centre de Calcul Scientifique en région Centre-Val de Loire. We also thank the "Plateforme d'Analyse Protéomique de Paris Sud-Ouest" (PAPPSO) for proteomic analysis.

## Author contributions

S.E.O., M.K.J., T.D.D.B., S.B., V.C. designed the research. E.A.S., I.C., V.V., E.L., M.D., R.P.D., T.T.D., H.J.J., B.L., M.D.B.B.L. and F.G.H acquired the data. E.A.S., I.C., C.C., H.J.J., A.O., C.J., C.B.W., A.L., N.G.G., N.P. R.P.D. S.E.O. M.K.J S.B. and V.C. analyzed the data. E.A.S., C.C., T.D.D.B., S.B. and V.C. wrote the article. All authors read and approved the final manuscript.

## Competing interests

R.D. and H.J. are CEO and CTO of Future Genomics Technologies, respectively. M.K.J. has a financial interest in Biomia. All other authors declare no competing interests.
