## [Peer Review File · Communications Biology]

Reviewers' comments:

Reviewer #1 & #2 (Remarks to the Author):

This manuscript investigates the production of yohimbane type monoterpene indole alkaloids (MIAs). In particular, the study utilises a combination of metabolomics/transcriptomics, genomics, proteomics, and machine learning to identify possible yohimbane synthases from *Rauvolfia tetraphylla*. In this combinatorial approach the authors have identified a yohimbane synthase of the medium chain dehydrogenase/reductase type and shed light upon the production of an important group of MIAs. Interestingly, the authors have also noted some potential yohimbane production activity in RtTHAS3, as well as RtVAS when in combination with RtGS and propose that there may be single or two enzyme production of yohimbane type MIAs. In the process of the work the authors have generated a good quality genome and transcriptomic resources for the community which will be made publicly available. Due to a lack of expertise, I cannot comment on the validity of the machine learning approach, or on the documentation of the methods used for this technique. The rest of the study is of good quality, novel, and robust. The authors have utilised a wide range of approaches to identify RtYOS, which will be of interest to the field, as well as proposing two interesting mechanisms. The data presented supports the conclusions and the appropriate controls have been included in all experimental data. However, a few minor improvements could be made to increase the transparency and readability of the manuscript:

1. [Line 210] the authors say that *R. tetraphylla* has an expansion of ADH enzymes compared to other plants. Supplementary Table 4 contains numerical data for just 4 species. Should the text read 'compared to other Gentianales?' Also, the table does not provide evidence for these genes being in the classes suggested. A phylogenetic tree would be better. Finally, is 'wide expansion' an overstatement? The numbers of the four classes are not consistently higher than the other three species shown.
2. [Figure 4] To aid interpretation, it would be very helpful to include the descriptive term for the GO categories in the figure rather than just the number. If it is impossible to fit both in the figure, then legend could include both? At present, the key messages of the data are not coming through.
3. [Figure 5] The legend boxes for each tissue type are somewhat small and the box plots seem slightly low resolution, this makes it a little difficult to see the colour associated with each tissue. The resolution may just be an issue with the reviewer documents, but it would improve the figure to increase the sizes of the boxes to make the colours clear.
4. In the discussion it would be helpful to clearly state which approach was most helpful for the identification of RtYOS. As best, I can tell MSTRG.5283.1 was identified based on the proteomics of latex and the machine learning model? It would be useful to have more clarity and detail on this.
5. [Lines 908-921] If permitted by journal guidelines, it would be helpful to include the link to the personalised script utilised (<https://doi.org/10.6084/m9.figshare.20749096.v1>) instead of, or in addition to, the paper it was used in previously.

Reviewer #3 (Remarks to the Author):

In the present study, the authors sequenced the genome of *R. tetraphylla* and identified the MIA related genes by integrating multi-omics and machine learning. The authors have conducted a comprehensive study with adequate bioinformatic analyses and in vitro validation. Despite a few issues, I find the study is well carried out and coherently presented and will broaden our understanding of the synthesis of plant bioactives.

Minor comments

The assembly results are fine, why not assemble the genome at chromosome-level by HiC sequencing results?

Line 264: The family name should not be italic.

Line 258: Please clarify the enrichment results of expanded and contracted genes. If MIA related genes in *R. tetraphylla* genome experienced expansion or contraction.

Line 780: I suggest to evaluate the assembly quality by using LAI index.

Line 807: I suggest to reconstruct phylogeny using RAXML or IQtree with the suitable substitution model.

Line 841 : Data were processed with the X! what's "X!"

Line 945: sixthree > sixteen??

Reviewer #4 (Remarks to the Author):

In the study titled, "The *Rauvolfia tetraphylla* genome suggests distinct biosynthetic routes for yohimbane monoterpene indole alkaloids", Stander et al. reported a newly assembled genome assembly for the medicinal plant, *Rauvolfia tetraphylla*, and used combination of transcriptome, genome and metabolome analysis to describe biosynthesis of yohimbanes. Authors identified and functionally characterized a medium-chain dehydrogenase/reductase, a multifunctional yohimbane synthase (YOS) involved in the biosynthesis of yohimbine and rauwolscine. Authors cloned 22 ADHs from *R. tetraphylla* and performed functional characterization by expressing them in yeast strains to explore genes involved in the biosynthesis and diversification of yohimbanes. The results described here are impressive. The authors have done a lot of hard work and have gone at lengths to look for all possibilities that might explain the biosynthesis of yohimbane in the target plant species. All experiments, experimental design and controls are on solid base. Results are extremely useful for advancing the understanding of MIAs biosynthesis, and authors have provided nice explanation in terms of their views on the evolution of MIAs in the context of yohimbane. I am impressed with this study, and support this article. I do have few suggestions that I am listing below-

1. Authors should provide more detail as how metabolome analysis was performed. The authors mention that a single Q was used. What conditions, manufacturers, LC conditions, MS conditions, column information, and other details are required here. Same comment for proteome analysis.
2. I suggest bringing Fig. 7 as a supplementary figure.
3. Please consider providing scripts and conditions in form of Git Hub for deep learning classification part, such that the entire results could be reproduced. Ideally, the gene lists used for training, and the script and version of R used for that training should be included. I feel that this is extremely important for better use of the datasets generated in this study. Also, please provide information on what computational system was used, single core? Method section lacks detail, and I recommend authors to provide these for readers to reproduce these results.
4. Line 909, "A personal script was used to identify regions of physically co-localized biosynthetic genes associated with MIA-annotated genes". I am assuming that the script is available through citation 17, but I think that it will be great if authors consider creating a github for this publication, and also include the script for gene cluster discovery in that repository. In my opinion, for an open and fair science, such resources are extremely important.
5. I recommend authors to check the manuscript carefully for potential errors in terms of formatting

and otherwise. Few places, statements are quite general and I disagree. For example, Line 70, "alkaloids are often found highly poisonous compounds" is a general statement and I think is not correct. Poisonous to whom? Again, line 73, "MIAs are also widespread...". Use of "also" is not correct here. Further, a plant, "Nothapodytes nimmoniana" from Icacinaceae also produces MIAs including Camptothecin. So, again, the statement is quite general and authors need to be careful to make such statement. Please correct this statement. Also, manuscript writing requires careful recheck.

All in all, this is an excellent study, and I am sure that it will help our understanding on MIA biosynthesis in plants. For these reasons, I support the publication of this article in Communication Biology.

- Reviewer #1 & #2 (Remarks to the Author):

1 [Line 210] the authors say that R. tetraphylla has an expansion of ADH enzymes compared to other plants. Supplementary Table 4 contains numerical data for just 4 species. Should the text read 'compared to other Gentianales?' Also, the table does not provide evidence for these genes being in the classes suggested. A phylogenetic tree would be better. Finally, is 'wide expansion' an overstatement? The numbers of the four classes are not consistently higher than the other three species shown.

Reviewers 1 and 2 are right. The ADH extension has been somehow overestimated, and the comparison was mainly focused on Apocynaceae. According to their comment, we modified L210 as follows :“Among the putative ADH genes, 79 MDR (PF00107.26, PF08240.12), 76 SDR (PF00106.25) and 27 AKR (PF00248.21) were identified in the R. tetraphylla genome suggesting a slight increase of these families compared to other Apocynaceae”. As recommended, we have also integrated an ADH phylogenetic tree (new Supplementary Figure S1) in the revised version besides keeping supplementary Tables.

2. [Figure 4] To aid interpretation, it would be very helpful to include the descriptive term for the GO categories in the figure rather than just the number. If it is impossible to fit both in the figure, then legend could include both? At present, the key messages of the data are not coming through.

Reviewer 1 and 2 are right. We have modified figure 4 to ease understanding by adding GO terms description and updated the legend in accordance with this comment. This new version has been included in the revised manuscript and below :

A

B

Response to stimuli

C

Specialized metabolism

3. [Figure 5] The legend boxes for each tissue type are somewhat small and the box plots seem slightly low resolution, this makes it a little difficult to see the colour associated with each tissue. The resolution may just an issue with the reviewer documents, but it would improve the figure to increase the sizes of the boxes to make the colours clear.

According to reviewer 1 and 2, the figure 2 has been redesigned at a higher resolution and by including color boxes of a higher size. All MIA quantifications are also available in the Supplemental Tables for a higher accuracy of the values.

4. In the discussion it would be helpful to clearly state which approach was most helpful for the identification of RtYOS. As best, I can tell MSTRG.5283.1 was identified based on the proteomics of latex and the machine learning model? It would be useful to have more clarity and detail on this.

According to reviewer 1 and 2 demand, we have included the following sentence in the conclusion to clarify enzyme identification “ While the use of a single prediction approach may result in the identification of interesting candidates such as YOS in the latex proteome, integration of several strategies including ML based approach, co-expression analysis and gene cluster screening first reinforced the aforementioned enzyme identification but also widened discovery to unexpected biosynthetic mechanism such as yohimbane synthesis through a double enzyme process.”

In addition, all methods leading to the identification of each candidate are available in Supplementary Table S14.

5. [Lines 908-921] *If permitted by journal guidelines, it would be helpful to include the link to the personalised script utilised (<https://doi.org/10.6084/m9.figshare.20749096.v1>) instead of, or in addition to, the paper it was used in previously.*

According to reviewers 1 and 2, the figshare URL has been included in the method section in addition to the reference of the corresponding article as follow : A personal script (accessible at <https://doi.org/10.6084/m9.figshare.20749096.v1>) was used to identify regions of physically co-localized biosynthetic genes associated with MIA-annotated genes [17].

- Reviewer #3 (Remarks to the Author):

1. *The assembly results fine, why not assemble the genome at chromosome-level by HiC sequencing results?*

We thank reviewer 3 for recognizing the quality of our assembly. Obtaining a HiC version of the genome would have been great but a bit out of the scoop of this work focused on enzyme identification.

2. *Line 264: The family name should not be italic.*

Reviewer 3 is right. Family names font has been corrected throughout the manuscript.

3. *Line 258: Please clarify the enrichment results of expanded and contracted genes. If MIA related genes in *R. tetraphylla* genome experienced expansion or contraction.*

We thank reviewer 3 for this remark. MIA-related genes are not present in expanded/contracted orthogroups. Therefore, we slightly modified our conclusions as follow: “Such evolutionary processes may account for the specialized metabolite variability found in closely related MIA-producing species, such as *R. tetraphylla* and *C. roseus*.”

4. *Line 780: I suggest to evaluated the assembly quality by using LAI index.*

According to reviewer 3 suggestion, we added LAI index results in Supplementary Table S1 and in the text L175 and L798:

“The base level QV of 32.6546, corresponding to more than 99,999% base accuracy, together with the LTR assembly index of 18.90 are a very good indicators of the high quality of the assembled genome.”

“Assembly quality assessment was performed combining the stat program from BBMap tool (v.38.94, [70]), MerQuiry (v. 1.3, [71]) and LTR assembly index from LTR_retriever (v2.9.6, [72]).”

5. *Line 807: I suggest to reconstruct phylogeny using RAXML or IQtree with the suitable substitution model.*

We are sorry for not having been clear enough in the first version of the manuscript. Indeed, the phylogenetic tree was obtained using raxml-ng option of Orthofinder. This has been specified in the revised version as follow:

“The resulting sequences were used as input for OrthoFinder (v.2.5.4; [87]) using the following parameters: -S diamond -M msa -A muscle -T raxml-ng.”

6. Line 841 : Data were processed with the X! what's “X!”

X!TandemPipeline is the pipeline used to analyze proteomics data. We agree with reviewer 3 that it is a surprising name. Reference to this pipeline was added [94]

“[94] Langella O, Valot B, Balliau T, Blein-Nicolas M, Bonhomme L, Zivy M. X!TandemPipeline: A Tool to Manage Sequence Redundancy for Protein Inference and Phosphosite Identification. *J Proteome Res.* 2017 Feb 3;16(2):494-503. doi: 10.1021/acs.jproteome.6b00632.”

7. Line 945: sixthree > sixteen??

Thanks for having noticed this mistake. We initially mean “six colonies of each strain”. This entire section has been rewritten to correct other typos. “To test the yeast strains expressing ADH candidates for production, six colonies of each strain were inoculated in 150 μ L synthetic complete medium lacking tryptophan (SC-Trp) and incubated overnight at 30 °C and 300 rpm in 96-well microtiter plates. After 16h, 5 μ L of each culture was transferred into 0.295 mL of SC-Trp supplemented with 0.25 mM secologanin and 1 mM tryptamine in deep well plates. The plates were then incubated for 96 h at 30 °C and 300 rpm. At the end of the culture, yeast were removed from the medium by ultrafiltration through an acroprep filter plate at 5000 g for 5 min. Supernatant was mixed with 9 volumes of methanol and 5 μ L was analyzed on UPLC-MS as described above in *Metabolite profiling of R. tetraphylla*.

- Reviewer #4 (Remarks to the Author):

1. Authors should provide more detail as how metabolome analysis was performed. The authors mention that a single Q was used. What conditions, manufacturers, LC conditions, MS conditions, column information, and other details are required here. Same comment for proteome analysis.

As recommended by reviewer 4, we have updated the description of the metabolomic analysis to include the missing information: “Metabolites were extracted from 94 different *R. tetraphylla* samples, obtained from 94 different tissue types and experimental conditions, described above in sections *Biotic interaction between R. tetraphylla* leaves and *RNA extraction and sequencing (Supplementary table S7)*. Samples were frozen in liquid nitrogen, then freeze-dried, ground into powder, and sonicated with methanol 0.1% of formic acid to extract metabolites. For *R. tetraphylla* latex, around 200 μ L of latex from freshly cut leaves were diluted in 200 μ L of PBS buffer and the alkaloids were extracted by adding one volume of methanol 0.1% of formic acid. After centrifugations, extracts were diluted in water Milli-Q 0.1% of formic acid and injected on a UPLC system (Acquity, Waters) coupled to a single quadrupole mass spectrometer with an 18-min linear gradient from 10 to 40% acetonitrile (containing 0.1% formic acid). Separation was performed using a Waters Acquity HSS T3 C18 column (150 mm*2.1 mm, 1.8 μ m) with a flow rate of 0.4 ml.min⁻¹ at 55°C, with an injection volume of 5 μ L. Mass spectrometry detection was performed using an SQD mass spectrometer

(SQD2, Waters), with capillary and sample cone voltages of 3 000 V and 30 V respectively, and cone and desolvation gas flow rates of 60 L.h⁻¹ and 800 L.h⁻¹. The selected mode of ion monitoring was employed in positive mode for the following compounds: ajmalicine (m/z 353), RT = 10.4; ajmaline (m/z 327), RT = 7.4; alstonine (m/z 349), RT = 11.7; corynanthine (m/z 355), RT = 7.9; isoreserpiline (m/z 413), RT = 9.0; loganic acid (m/z 377), RT = 2.1; rauwolscine (m/z 355), RT = 6.3; reserpiline (m/z 413), RT = 10.1; serpentine (m/z 349), RT = 11.6; strictosidine (m/z 531), RT = 9.0; tetrahydroalstonine (m/z 353), RT = 10.6; tryptamine (m/z 161), RT = 3.0; vinorine (m/z 335), RT = 8.6; yohimbine (m/z 355), RT = 7.7.

In addition, the whole description of the proteomic analysis has been appended to the material and method section of the revised manuscript has follow:

Protein in gel digestion

Gel pieces were washed thrice by successive separate baths of 10% acetic acid, 40% ethanol and 25 mM ammonium bicarbonate and ACN. Proteins were reduced for 30 min with 10 mM dithiothreitol, 25 mM ammonium bicarbonate at 56 °C and alkylated in the dark with 55 mM iodoacetamide, 25 mM ammonium bicarbonate for 45 min at room temperature. The gel pieces were washed by successive separate baths of 0.1 M ammonium bicarbonate and ACN. Digestion was subsequently performed overnight at 37°C with 125 ng of modified trypsin (Promega) dissolved in 50 mM ammonium bicarbonate. The peptides were extracted successively with 0,5% trifluoroacetic acid (TFA) and 50% ACN and then with ACN. Peptide extracts were dried in a vacuum centrifuge.

LC MS/MS analysis

Peptide samples were solubilized in a buffer of 2% CH₃CN, 0.1% formic acid. Liquid chromatography was performed on a NanoLC Ultra system (Eksigent, Dublin, CA, USA). Samples were loaded at 7.5 µl min⁻¹ on a C18 precolumn (5 µm, 100 µm i.d. × 2 cm length; NanoSeparations) connected to a separating BIOSPHERE C18 column (3 µm, 75 µm i.d. × 300 mm length; NanoSeparations). Solvent A was 0.1% formic acid in water and solvent B was 0.1% formic acid in CH₃CN. Peptide separation was achieved using a linear gradient from 5 to 35% solvent B for 28 min at 300 nl min⁻¹. Including the regeneration and the equilibration steps, a single run took 45 min. Eluted peptides were analyzed with a Q Exactive™ Plus mass spectrometer (Thermo Fisher Scientific) using a nanoelectrospray interface. Ionization was performed with a 1.3 kV spray voltage applied to an uncoated capillary probe (10 µm i.d., New Objective). The Xcalibur interface was used to monitor data-dependent acquisition of peptide ions. This included a full MS scan covering a mass-to-charge ratio (m/z) of 350 to 1400 with a resolution of 70 000 and an MS/MS step (normalized collision energy, 27%; resolution, 17 500). The MS/MS step was reiterated for the eight major ions detected during the full MS scan. Dynamic exclusion was set to 50 sec. Only doubly and triply charged precursor ions were subjected to MS/MS fragmentation.

Identification of peptides

A database searches were performed using X!Tandem Alanine (Release 2017.2.1.4) (<http://www.thegpm.org/TANDEM>). Enzymatic cleavage was described to be due to trypsin digestion with one possible misscleavage. Cys carboxyamidomethylation was set as static modification whereas Met oxidation, Nter deamidation and Nter acetylation were set as variable modifications. Identifications were performed using a user-supplied database and an

internal database of standard contaminants (trypsin, keratins, BSA). Identified proteins were filtered and grouped using X!TandemPipeline v0.2.38 (<http://pappso.inrae.fr/bioinfo/i2masschroq/>) [94]. Data filtering was achieved according to a peptide E value smaller than 0.01 with a minimum of 2 peptide to identify a protein.

2. I suggest bringing Fig. 7 as a supplementary figure.

We thank reviewer 4 for this recommendation. If possible, we would prefer keeping this figure in the manuscript given the presence of YOS in the latex.

3. Please consider providing scripts and conditions in form of Git Hub for deep learning classification part, such that the entire results could be reproduced. Ideally, the gene lists used for training, and the script and version of R used for that training should be included. I feel that this is extremely important for better use of the datasets generated in this study. Also, please provide information on what computational system was used, single core? Method section lacks detail, and I recommend authors to provide these for readers to reproduce these results.

Reviewer 4 is right, and we recently published an article entirely describing the script for deep learning. This new reference, as well as algorithm training parameters and computational system, have been included in the revised version of the manuscript as follow: “To predict the MIA-related genes from our gene expression atlas with ANN, we referred to our recently described protocol (Dugé de Bernonville et al., 2022). The input dataset was the same than the one used to construct co-expression network. The H2O library (v3.36) in R (v4.2) was used to train a feedforward neural network with backpropagation. To define the true positive events, a number of genes were labeled as MIA as described above. For the true negative events (non-MIA related genes), we used genes predicted as conserved orthologs with the BUSCO tool. Because these genes are strongly conserved across very diverse plant families, they are not expected to belong to the MIA pathway. We finally labeled 75 genes as MIA related and 1908 as non-MIA related. This set was split into a training and a validation dataset following a 70/30 partition with seed 666. Once a network (seed 666) was trained, we looked at the number of MIA predicted genes using the full dataset (with data unseen by the network). The logloss was used as a stopping metric. A hyperparameter search revealed that a simple architecture containing 1 input layer (57 samples), 2 hidden layers (with 40 and 20 neurons, respectively) and an output layer containing 2 values (MIA vs non-MIA) provided the most relevant predictions. We added dropout ratios (0.1 in the input and 0.5 in each hidden) to improve the model generalization. The number of epochs was selected by visually inspecting the logloss progression during iterations. A value of 1,500 was found to be the most appropriate to avoid overfitting. Calculations were done on a classical computer device (8 CPU, RAM 16 Gb)”

In addition, all the data used for training of the algorithm have been included in the manuscript in the different supplement tables. To easily reproduce this analysis and the random draw, we have also specified the see parameter (666 for both cases).

Finally, we have also included the Git-hub URL in the CODE AVAILABILITY section of the revised manuscript (<https://github.com/EA2106-Universite-Francois-Rabelais>)

4. Line 909, “A personal script was used to identify regions of physically co-localized biosynthetic genes associated with MIA-annotated genes”. I am assuming that the script is available through citation 17, but I think that it will be great if authors consider creating a github for this publication, and also include the script for gene cluster discovery in that repository. In my opinion, for a open and fair science, such resources are extremely important.

Reviewer 4 is right, the personal script used to identify those regions is described in reference 17. In addition, to ease access and use of this resource, we have also provided the figshare URL for direct access to this script: A personal script (accessible at <https://doi.org/10.6084/m9.figshare.20749096.v1>) was used to identify regions of physically co-localized biosynthetic genes associated with MIA-annotated genes [17].

5. I recommend authors to check the manuscript carefully for potential errors in terms of formatting and otherwise. Few places, statements are quite general and I disagree. For example, Line 70, “alkaloids are often found highly poisonous compounds” is a general statement and I think is not correct. Poisonous to whom? Again, line 73, “MIAs are also widespread...”. Use of “also” is not correct here. Further, a plant, “Nothapodytes nimmoniana” from Icacinaceae also produces MIAs including Camptothecin. So, again, the statement is quite general and authors need to be careful to make such statement. Please correct this statement. Also, manuscript writing requires careful recheck.

We agree with reviewer 1 that we used general characteristics to describe monoterpene indole alkaloids in the introduction section. These are the accepted ones found in the most recent articles. To gain in accuracy and in agreement with reviewer 4 concerns, we have updated the corresponding sentences as follow:

For example, Line 70, “alkaloids are often found highly poisonous compounds” is a general statement and I think is not correct. **This sentence has been changed into “alkaloids are compounds with diverse physiological roles and often described as highly poisonous molecules that protect plants against pests and herbivores.” The fact that many MIAs display toxicity against bioagressors is well known and accepted but to minor this role, we have first talked about diverse physiological roles.**

Again, line 73, “MIAs are also widespread...”. Use of “also” is not correct here. Further, a plant, “Nothapodytes nimmoniana” from Icacinaceae also produces MIAs including camptothecin. **This sentence has been changed into “MIAs are widespread within the Gentianales order including Apocynaceae, Gelsemiaceae, Loganiaceae, and Rubiaceae, and are also biosynthesized in Nyssaceae (Cornales order) and in Icacinaceae “**

Finally, the whole manuscript has been carefully rechecked.

REVIEWERS' COMMENTS:

Reviewer #1 (Remarks to the Author):

All reviewer comments have been addressed with care and the manuscript revised appropriately. I have no additional suggestions.

Reviewer #4 (Remarks to the Author):

The updated version of manuscript is significantly improved. I am satisfied with the responses that authors have given for my comments and the changes that they have made in the manuscript. I now endorse the manuscript for publication.